# Non-Invertible Defects in Nonlinear Sigma Models and Coupling to Topological Orders

Po-Shen Hsin[1]

[1] *Mani L. Bhaumik Institute for Theoretical Physics, Department of Physics and Astronomy, University of California, Los Angeles, CA 90095, USA*

**Abstract**

Nonlinear sigma models appear in a wide variety of physics contexts, such as the long-range order with spontaneously broken continuous global symmetries. There are also large classes of quantum criticality admit sigma model descriptions in their phase diagrams without known ultraviolet complete quantum field theory descriptions. We investigate defects in general nonlinear sigma models in any spacetime dimensions, which include the "electric" defects that are characterized by topological interactions on the defects, and the "magnetic" defects that are characterized by the isometries and homotopy groups. We use an analogue of the charge-flux attachment to show that the magnetic defects are in general non-invertible, and the electric and magnetic defects form junctions that combine defects of different dimensions into analogues of higher-group symmetry. We explore generalizations that couple nonlinear sigma models to topological quantum field theories by defect attachment, which modifies the non-invertible fusion and braiding of the defects. We discuss several applications, including constraints on energy scales and scenarios of low energy dynamics with spontaneous symmetry breaking in gauge theories, and axion gauge theories.

December 29, 2022

# 1 Introduction

Nonlinear sigma models appear a wide variety of physics context, such as in Ginzburg–Landau paradigm of phase transitions characterized by spontaneously broken continuous global symmetries, where the corresponding Nambu-Goldstone modes are described by the sigma model associated with the broken symmetries [1, 2]. A large class of quantum criticality such as [3, 4, 5] can also be described by nonlinear sigma models as part of the phase diagram, while the description for the entire phase diagram of the quantum criticality often remains unclear.

Defects play important role in probing phases and phase transitions, such as defect-driven phase transitions [6, 7, 8], and constraining the renormalization group flows using the correlation function of topological defects. For instance, the correlation functions of the topological defects that generate one-form symmetry constrains confinement in gauge theories, and the correlation functions involving topological defects generating the chiral symmetry constrain the chiral symmetry breaking at low energy (see *e.g.* [9]). Defects also impose consistency conditions on the phases and phase transitions across the entire phase diagram [10] (see also [11, 12]). Topological defects also lead to selection rules in correlation functions, and defects that are approximately topological can provide possible solutions to naturalness and hierarchy problems (see *e.g.* [13, 14]). The violation of the topological

condition for defects can also probe the particle spectrum in the theory related to the conjectures in quantum gravity (see *e.g.* [15]). Topological defects in suitable lattice models can also realize logical gates in quantum codes on the ground state subspace in the Hilbert space, see *e.g.* [16, 17, 18, 19].

In this note, we investigate the properties of defects in general nonlinear sigma models in any spacetime dimension. We study two classes of defects, which we call the electric and the magnetic defects. The electric defects are defined by topological interactions on the defect. The magnetic defects are better known in the literature, see *e.g.* [20]. They are specified by the boundary conditions around the defect. For instance, the magnetic defects of codimension three and higher can be specified by the degree two and higher homotopy groups of the target space of the nonlinear sigma model, which are always Abelian groups. Nevertheless, we discovered that in the presence of topological interactions, the magnetic defects can obey non-invertible fusion rules and non-Abelian braiding statistics. Such non-invertible defects are discussed in many gauge theories and lattice models, see *e.g.* [21, 22], whose low energy dynamics are often unknown. Thus it is important to understand these defects in the possible scenarios for the low energy dynamics, such as spontaneously broken continuous symmetry described by nonlinear sigma models.

There are gauge theories with conjectured spontaneously broken chiral symmetry and topological order, where the low energy dynamics can be described by nonlinear sigma models coupled to topological quantum field theories (TQFTs). To understand the renormalization group flows in such theories, it is important to investigate the couplings between general sigma models and TQFTs. We study the coupling between general nonlinear sigma model and TQFT by starting with a continuous family of gapped systems with the same topological order, as studied in *e.g.* [12, 10], and then promote the parameters that label the family to be the dynamical sigma model fields. We show that the couplings correspond to modification on the defects, such as modifying the fusion and braiding relations.

The note is organized as follows. In Section 2, we discuss the "electric" and "magnetic" defects in general nonlinear sigma model protected by the topology of the target space. In Section 3, we discuss the generalization of the charge-flux attachment for the electric and magnetic defects induced by topological interactions, and argue that the defects form analogues of higher-group structures. In Section 4, we show the topological interaction can make the magnetic defects non-invertible. In Section 5, we study the correlation functions of the defects. In Section 6, we discuss the generalizations that couple sigma models to topological quantum field theories. In Section 7, we discuss examples of symmetry matching between the ultraviolet (UV) and the infrared (IR) in dynamics scenarios with spontaneously broken continuous symmetries. In Section 8, we discuss another class of examples of gauge theories with axions in 3+1D, where we found that there are defects that obey three-loop braiding relations, fermionic string statistics, and non-Abelian braiding.

## 2 Defects in nonlinear sigma model

Let us consider sigma model on $D$-dimensional spacetime $X$, where the sigma model field $\lambda$ takes value in the target space $M$. The sigma model field $\lambda$ is a map

$$\lambda: \quad X \to M \ . \tag{2.1}$$

For now, we will consider sigma models with no interactions other than the usual kinetic term given by the metric on $M$ and the topological actions.

## 2.1 "Electric" defects: topological terms on submanifolds

The sigma model has various "electric" defects given by topological terms of the sigma model field.[1] Denote the topological action by $S_{\text{top}}^{(n)}[\lambda, M_n]$ on an $n$-dimensional submanifold $M_n$, the insertion of the defect can be expressed as modifying the path integral:

$$Z = \int D\lambda e^{iS_{\text{kinetic}}[\lambda]} e^{iS_{\text{top}}^{(n)}[\lambda, M_n]} \ . \tag{2.2}$$

When $n = D$ and $M_D = X$, such spacetime-filling electric defect represents a topological action of the sigma model.

Let us describe three classes of electric defects (there can be overlap between these classes), with topological action given by

(1) $\eta^{(n)} \in H^n(M, U(1))$. The topological action is given by

$$S_{\text{top}}^{(n)}[\lambda, M_n] = \int_{M_n} \lambda^* \eta^{(n)} \ , \tag{2.3}$$

where $\lambda^* \eta^{(n)}$ denotes the pullback of $\eta^{(n)}$ by $\lambda : X \to M$.

(2) $\zeta^{(n)} \in H^{n+1}(M, \mathbb{Z})$. The action is written in terms of an auxiliary $(n+1)$-dimensional manifold $V_{n+1}$ whose boundary is $M_n$,

$$S_{\text{top}}^{(n)}[\lambda, M_n] = \int_{V_{n+1}} \lambda^* \zeta^{(n)} \ , \tag{2.4}$$

and it is independent of the choice of the bounding manifold $V_{n+1}$. In some cases (or under suitable definition of cohomology), they coincide with the defects in class (1).[2]

Such defects also represent Wess-Zumino terms on submanifolds.

(3) Theta term $\theta^{(n)} \in H^n(M, \mathbb{Z})$. The topological action is given by the theta term

$$S_{\text{top}}^{(n)}[\lambda, M_n] = \alpha \int_{M_n} \lambda^* \theta^{(n)} \ , \tag{2.5}$$

where $\alpha \in \mathbb{R}/2\pi\mathbb{Z}$ is an angular parameter. Such defect is topological and always admits a boundary.

---

[1]They are the analogues of the antisymmetric two-form $B$ field in String theory.

[2]The defects in (1),(2) are related by the connecting homomorphism $H^n(M, U(1)) \to H^{n+1}(M, \mathbb{Z})$ in the long exact sequence $\cdots \to H^n(M, \mathbb{R}) \to H^n(M, U(1)) \to H^{n+1}(M, \mathbb{Z}) \to H^{n+1}(M, \mathbb{R}) \to \cdots$ for the short exact sequence $\mathbb{Z} \to \mathbb{R} \to U(1)$.

### 2.1.1 Examples of electric defects: $M = S^1, S^2, BG$

Let us describe the electric defects for some examples of target space $M$.

- Example: $M = S^1$. Denote the sigma model field by $\lambda \sim \lambda + 2\pi$. There are theta term type electric line defects described by $e^{i\alpha \int_{M_1} d\lambda/2\pi}$. If we regard the sigma model as the Nambu-Goldstone boson for spontaneously broken $U(1)$ symmetry, the electric line defect corresponds to the boson particles. The electric line defect can end at the point $e^{i\alpha\lambda/2\pi}$.

- Example: $M = S^2$. There are theta term type electric surface defect $\theta^{(2)} \in H^2(S^2)$ and the electric line defect $\zeta^{(1)} \in H^2(S^2)$. The surface defect is $e^{i\alpha \int_{M_2} \lambda^* \omega_2}$ where $\omega_2$ is an integer multiple of the volume two-form on $S^2$ with integral one. The electric line defect is $e^{i \int_{M_1} \lambda^* \tau_1}$ where $\tau_1$ is an integer multiple of the Berry connection on $S^2$.

  If we regard $S^2 = SU(2)/U(1)$ as gauging $U(1)$ subgroup isometry symmetry in $SU(2)$ sigma model, then the electric line defects are the Wilson lines of the $U(1)$ gauge field, and the electric surface defect is theta term of the $U(1)$ gauge field.

- Example: $M = BU(1)$ the classifying space of $U(1)$, which is the same as pure $U(1)$ gauge theory. There are electric Wilson lines $H^2(BU(1), \mathbb{Z}) = \mathbb{Z}$, and there are theta term type electric surface defect. In terms of the $U(1)$ gauge field $a$, they are $e^{iq_e \oint_{M_1} a}$ and $e^{i\alpha n \int da/2\pi}$ where $n$ is an integer.

- Example: $M = BG$ for some group $G$, such sigma model is the same as pure gauge theory with gauge group $G$. The electric defects are given by defects that supported on submanifolds decorated with gauged symmetry-protected topological (SPT) phases of the $G$ gauge fields [19].

## 2.2 "Magnetic" defects

The sigma model has various "magnetic defects", specified by boundary conditions around the defect. The insertion of magnetic defect is equivalent to modifying the path integral over $\lambda$ that has fixed classical configuration around the magnetic defect. These defects were discussed in *e.g.* [20]. The magnetic defects of codimension $k$ can be surrounded by a $(k-1)$-dimensional sphere, and the magnetic defects are characterized by the map $S^{k-1} \to M$ that gives the boundary condition of the sigma model field around the defect.

Let us describe four classes of magnetic defects:

(1) Magnetic defect of codimension one. We can sandwich the domain wall with two points, and map the first point to a reference point on $M$, while the other point maps to the image under a homeomorphism $\rho : M \to M$. Thus the codimension-one magnetic defects correspond to the homeomorphisms on $M$.

   The topological magnetic defects are those that preserve the kinetic term, which is given by the metric on $M$. Thus the topological codimension-one magnetic defects correspond to the isometries on $M$.[3]

---

[3]As we will discuss in Section 3, in the presence of a topological action $\omega^{(D)}$ for the sigma model, the isometries of $M$ that change the topological action still represent topological codimension one defect, but the defect becomes

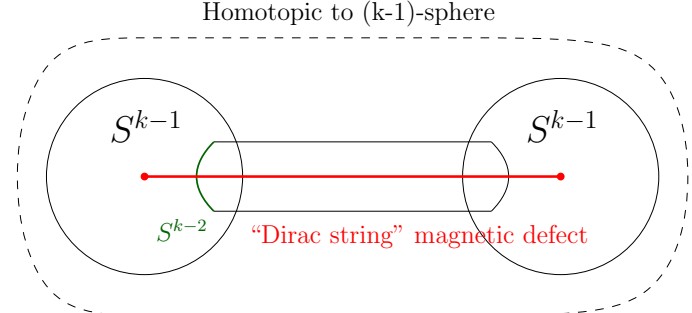

Figure 1: There are magnetic defects of codimension $(k-1)$ that are the analogues of "Dirac strings" for improperly quantized magnetic defects of codimension $k$, for very short "Dirac string" this is a properly quantized magnetic defect of codimension $k$.

For the magnetic defects given by isometries $m, m'$, we define $m \circ m'$ to be the magnetic defect using the group multiplication of the isometry group of $M$.

(2) Magnetic defect of codimension two. We can surround the magnetic defect of codimension two by a circle, and the defect is characterized by the conjugacy classes of $\pi_1(M)$ [20]. We will focus on the magnetic defects that correspond to the center conjugacy class of $\pi_1(M)$.

For the magnetic defects $[m], [m'] \in Z(\pi_1(M))$ (where $Z(\cdot)$ denotes the center of group $\cdot$), we define $m+m'$ to be the representative map from $S^1 \to M$ for the element $[m]+[m'] \in Z(\pi_1(M))$.

(3) Magnetic defect of codimension $k \geq 3$. These defects are characterized by $\pi_{k-1}(M)$, and they are not topological.

For the magnetic defects $[m_k], [m'_k] \in \pi_{k-1}(M)$, we define $m_k + m'_k$ to be the representative map from $S^{k-1} \to M$ for the element $[m_k] + [m'_k] \in \pi_{k-1}(M)$.

(4) "Theta term" or "Dirac string" type magnetic defects of codimension $k$, labelled by integral classes in $\pi_{k-1}(M)$. These defects are the generalizations of the Dirac string ending on improperly quantized monopole.[4]

These "Dirac string" defects are defined as follows: we first consider codimension-$k$ magnetic defect surrounded by $S^{k-1}$, then we elongate the $(k-1)$-dimensional sphere into two bulbs connected by a thin tube. Each bulb can be regarded as $S^{k-1}$ with a hole removed, and the boundary of the hole surrounds the codimension-$(k-1)$ Dirac string. Due to the removal of the hole, we need to specify a map on the $(k-1)$-sphere with the hole removed with additional boundary condition around the $(k-2)$-sphere surrounding the hole. See Figure 1 for an illustration.

### 2.2.1 Examples of magnetic defects: $M = S^1, S^2, BG$

Let us describe the electric defects for some examples of target space $M$.

---

non-invertible.

[4]The reason that we focus on the Dirac strings for integer classes is that there is a density that we can integrate over the manifold with boundary given by $S^{k-1}$ with a hole removed to define improperly quantized monopole.

- Example: $M = S^1$. There are codimension-one magnetic defect given by $O(2)$ isometry on $S^1$. There are codimension-two magnetic defect given by $\pi_1(S^1) = \mathbb{Z}$. The codimension-one theta term type magnetic defects coincide with the defects that generate $U(1) \subset O(2)$ isometry of $M$.

  If we regard $S^1$ sigma model as the Nambu-Goldstone boson from spontaneously broken $U(1)$ symmetry, then the magnetic codimension two magnetic defect describes the vortex around which the phase of order parameter winds.

  If we regard $S^1$ sigma model as the axion, then the magnetic defects of codimension two are the axion strings.

- Example: $M = S^2$. There are codimension-one magnetic defects given by $O(3)$ isometry of $M$. There are codimension-three magnetic defect given by $\pi_2(S^2) = \mathbb{Z}$, magnetic defects of codimension four given by $\pi_3(S^2) = \mathbb{Z}$, and magnetic defects of codimension $k \geq 5$ given by $\pi_{k-1}(S^2)$. There are theta term type magnetic defects of codimension two and three.

  If we regard $S^2 = SU(2)/U(1)$ as gauging $U(1)$ symmetry in $SU(2)$ sigma model, the magnetic defects of codimension three are the monopole of the $U(1)$ gauge field, the theta term type magnetic defect of codimension two is the magnetic defect that carries $U(1)$ holonomy around it.

- Example: $M = BU(1)$, which is the same as pure $U(1)$ gauge theory. There are magnetic defect of codimension three $\pi_2(BU(1)) = \pi_1(U(1)) = \mathbb{Z}$, they are the monopoles. There are theta term type magnetic defect of codimension two, and they carry $U(1)$ holonomy around them.

- Example: $M = BG$, which is the same as pure gauge theory with gauge group $G$. There are magnetic defects of codimension one given by the automorphism of $G$. There are magnetic defects of codimension two, given by the conjugacy classes of $\pi_1(BG) = \pi_0(G)$. Similarly, there are magnetic defects of codimension $k \geq 3$, given by $\pi_{k-1}(BG) = \pi_{k-2}(G)$. There can also be theta term type magnetic defects.

## 2.3 Defects stuck at junctions

We can form higher-codimensional junctions of the defects discussed above, where multiple defects meet. When all these constituent defects are topological, the higher-codimensional junction is also topological. When each of the constituent defects cannot have boundaries, the junction also cannot have boundaries.

We remark that if the properties of the defects at the higher-codimensional junction does not correspond to any isolated defects, then such higher-codimensional junctions cannot be replaced by existing defects and are intrinsic properties of the constituent defects. Different consistent modifications of the junctions by stacking the junction with isolated defects correspond to different "symmetry fractionalizations", see *e.g.* [23, 24, 25, 26, 27, 28, 29, 30, 31, 32, 33].

### 2.3.1 Topological magnetic defects at junctions

The magnetic defects in sigma models are often non-topological, with the exceptions of the codimension-one magnetic defects given by the isometry. We can use them to construct higher-codimensional topological magnetic defects stuck at the junctions of the codimension-one magnetic defects.

The codimension-one magnetic defects generate isometry on $M$, and thus junctions of codimension-one magnetic defect can realize higher-codimensional defect that has boundary condition with winding number on $M$. Moreover, since the codimension-one isometry defects are topological (they might not be invertible, as we will discuss in Section 4), such higher-codimensional junctions are also topological. While isolated higher-codimensional magnetic defects may not be topological, such higher-codimensional magnetic defects stuck at the junctions of codimension-one defects are always topological, and they cannot have boundaries unless the codimension-one defects can have boundaries.

Denote the isometry group on $M$ by $\text{Isom}(M)$. It gives an action $\rho : \text{Isom}(M) \times M \to M$. The codimension-$k$ junctions of the codimension-one isometry defects can be characterized by $f_k \in \pi_{k-1}(\text{Isom}(M))$.

For instance, if $\text{Isom}(M) = SO(N) = Spin(N)/\mathbb{Z}_2$, the codimension-two junction characterized by non-trivial $w_2^{SO} \in H^2(BSO(N), \mathbb{Z}_2)$ corresponds to the composition of isometries that form a non-contractible one-cycle in $SO(N)$ (i.e., it is the non-trivial element of $\pi_1(SO(N)) = \mathbb{Z}_2$), which lifts to a non-closed path in $Spin(N)$ with endpoints identified by the $\mathbb{Z}_2$ quotient in $Spin(N)/\mathbb{Z}_2 = SO(N)$.

The homotopy group $\pi_{k-1}(\text{Isom}(M))$ gives a family of isometries over $S^{k-1}$, $f_k : S^{k-1} \to \text{Isom}(M)$. The magnetic defect stuck at the codimension-$k$ junction is given by

$$\widetilde{m}_k = \rho \circ f_k : \quad S^{k-1} \to M , \tag{2.6}$$

where we omitted a tensor product with the identity map.[5]

### 2.3.2 Example: $M = S^2$

Consider the isometry $SO(3)$ of $M = S^2$ sigma model, which has $H^2(BSO(3), U(1)) = \mathbb{Z}_2$. The trivalent junctions of the codimension-one isometry defects $g_1, g_2, g_1 g_2$ is characterized by the generator $w_2^f \in H^2(BSO(3), U(1))$ with $w_2^f(g_1, g_2) \in \mathbb{Z}_2$, which equals the even/odd parity of the homotopy class of $g_1 \circ g_2 \circ (g_1 g_2)^{-1} : S^2 \to S^2$.

When the homotopy class for such map is odd, the codimension-two junction of the isometry defects is the topological version of the Dirac string type defect for $\pi_2(S^2)$. In particular, it has $\pi$ mutual statistics with the electric line defects of the odd classes in $H^2(S^2, \mathbb{Z}) = \mathbb{Z}$. In terms of $S^2 = SU(2)/U(1)$ as gauging $U(1)$ symmetry in $SU(2) = S^3$ sigma model, the electric line defects are the $U(1)$ Wilson lines, and such braiding means that the $SO(3)$ isometry acts projectively on the odd charge Wilson lines, i.e. the odd charge Wilson lines carry half-integer isospin projective representations of the $SO(3)$ isometry symmetry.

---

[5]We note that the resulting map may belong to the trivial homotopy class in $\pi_{k-1}(M)$, but as long as the homotopy to the trivial map cannot be expressed as $\rho \circ f_k'$ for some homotopy $f_k'$ between $f_k$ and the trivial map, the magnetic defect stuck at the junction is nontrivial.

We note that there are also isolated Dirac string magnetic defects that braid with the electric line defects, but the electric line defects can end (in the $S^2 = SU(2)/U(1)$ presentation, the $U(1)$ gauge field couples to electric matter), and thus such isolated Dirac string magnetic defect is not topological, unlike the topological magnetic defects of codimension two stuck at the junctions of the isometry defects. (It is consistent for such topological junction to braid with the electric line defect that can end, since the isometry acts on the end points of the electric line defects.)

# 3 "Higher-group" junction from charge-flux attachment

In this section, we will show that the magnetic defects and electric defects form higher-group like junction. We begin by showing that topological action of the sigma model fields, which we denote by $\omega^{(D)}$, implies that the magentic defects are attached to electric defects.

## 3.1 Topological interactions attach electric defects to magnetic defects

We will show that in the presence of topological action $\omega^{(D)}$ for the sigma model with target space $M$ in $D$ dimensional spacetime, the magnetic defects are attached to electric defects by an analogue of the charge-flux attachment or the Witten effect.

We will discuss separately the case of the magnetic defects with codimension one, which is more obvious, and the case of the magnetic defects with higher codimensions. For the magnetic defect of higher codimensions, we will use similar arguments as in [19] (see also [34]).

### 3.1.1 Codimesion-one magnetic defects

Consider codimension-one magnetic defect that corresponds to the isometry $\rho$ on the target space $M$. Suppose the sigma model has topological action $\omega^{(D)}$ in $D$-dimensional spacetime. Then the action of the isometry $\rho : M \to M$ induces a permutation action on the possible topological actions $H^D(M, U(1)) \to H^D(M, U(1))$ (and also $H^{D+1}(M, \mathbb{Z}) \to H^{D+1}(M, \mathbb{Z})$). Thus it changes the topological action $\omega^{(D)}$ to $\rho\omega^{(D)}$. The codimension-one magnetic defect thus attaches to the $D$-dimensional defect

$$e^{i \int_{M_D} \lambda^*(\rho\omega^{(D)} - \omega^{(D)})} , \tag{3.1}$$

on a $D$-dimensional submanifold $M_D$ that bounds the magnetic defect.

The discussion here and below can be generalized to other types of electric defects in Section 2.1 such as Wess-Zumino terms by replacing $H^D(M, U(1))$ with $H^{D+1}(M, \mathbb{Z})$.

### 3.1.2 Higher-codimension magnetic defects

Let us consider the spacetime to be locally $\mathbb{R}^k \times \mathbb{R}^{D-k}$ and elongate $\mathbb{R}^k$ to a cigar geometry described by fibration of $S^{k-1}$ over $[0, \infty)$. We insert a codimension-$k$ magnetic defect at the tip of the cigar. (See Figure 2.) Then by reducing the theory over $S^{k-1}$ we find that the magnetic defect of codimension $k$ is attached to an electric defect of codimension $(k-1)$, given by the integration of the topological action along the fiber $S^{k-1}$ with the holonomy prescribed by the magnetic defect. When $k = 2$, we will focus on the magnetic defects correspond to the center of $\pi_1(M)$.

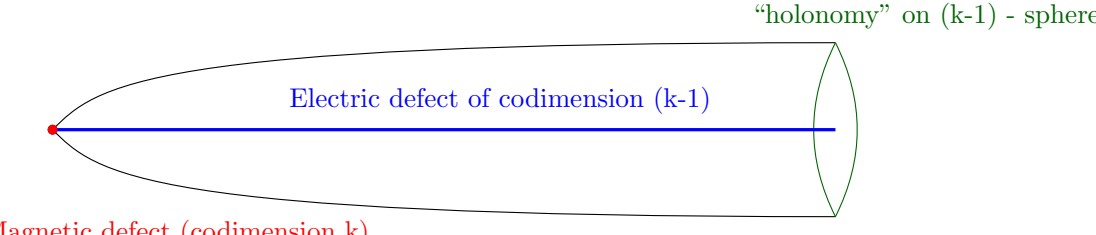

Figure 2: Magnetic defect of codimension $k$ is attached to electric defect of codimension $(k-1)$ given by the integration of the topological interaction $\omega^{(D)}$ over the $S^{k-1}$ fiber with background configuration specified by the magnetic defect $m_k : S^{k-1} \to M$ for the target space $M$. The emitted electric defect is computed by the product $i_{m_k}\omega^{(D)}$.

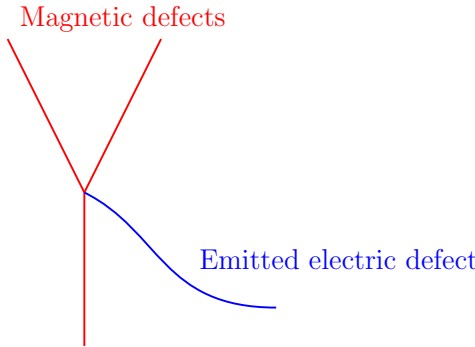

Figure 3: In the presence of topological action, the trivalent junction of magnetic defects emits an electric defect.

For the topological action $\omega^{(D)}$ that is a representative cocycle for an element in $H^D(M, U(1))$, and the magnetic defect $[m] \in \pi_{k-1}(M)$, such fiber integration gives the electric defect that attaches to the magnetic defect computed by the cap product, and we denote the result by $i_m\omega^{(D)}$:

$$i_m\omega^{(D)} \equiv h(m) \cap \omega^{(D)} \ , \tag{3.2}$$

where $h$ is the Hurewicz homomorphism [35] that sends a representative $(k-1)$-cycle of the generator of $H_{k-1}(S^{k-1})$ to an integral $(k-1)$-cycle on $M$ using the map $m : S^{k-1} \to M$.

Similar to the discussions in [19], we will decompose the result $i_m\omega^{(D)}$ into two parts: one part implies that the trivalent junction of the magnetic defects emits an electric defect, while the other part attach the magnetic defect to an electric defect that is an non-trivial element in $H^{D-k+1}(M, U(1))$. Suppose $[\omega^{(D)}] \in H^D(M, U(1))$ has order $N$, we can express it as $\omega^{(D)} = \frac{2\pi}{N}\omega^{(D);N}$ for $\omega^{(D);N} \in H^D(M, \mathbb{Z}_N)$. The class $[i_m\omega^{(D)}] \in H^{D-k+1}(M, U(1))$ has order $N'|N$, and we can choose a cocyle representative $\frac{2\pi}{N'}x$. Then $i_m\omega^{(D),N} = \frac{N}{N'}x + \frac{1}{N}dy$ for some $\mathbb{Z}_N$ $(D-k)$-cocycle $y$ on $M$. Denote the $(D-k+1)$-cocycle $i_m^A\omega^{(D)} = \frac{2\pi}{N'}x$ and the $(D-k)$-cocycle $i_m^B\omega^{(D)} = \frac{2\pi}{N}dy$, we have the decomposition

$$i_m\omega^{(D)} = i_m^A\omega^{(D)} + \frac{1}{|\omega^{(D)}|}di_m^B\omega^{(D)} \ . \tag{3.3}$$

where $|\omega^{(D)}| = N$ denotes the order of $[\omega^{(D)}] \in H^D(M, U(1))$. The part $i^B \omega^{(D)}$ modifies the trivalent junction of magnetic defects: at the junction of the magnetic defect $m, m', m + m'$, there emits the codimension-$k$ electric defect

$$\Omega_{m,m'}\omega^{(D)} \equiv \frac{1}{|\omega^{(D)}|} \left( i_m^B \omega^{(D)} + i_{m'}^B \omega^{(D)} - i_{m+m'}^B \omega^{(D)} \right) . \tag{3.4}$$

To summarize, the effect of topological interaction $\omega^{(D)}$ is the following:

- The magnetic defect $m$ is attached to an electric defect of one dimension higher as in Figure 2, given by $[i_m \omega^{(D)}]$.

- The trivalent junction of the magnetic defects $m, m', m + m'$ emits an electric defect as in Figure 3, given by (3.4).

The discussion can be generalized to other types of defects by replacing $H^D(M, U(1))$ with $H^{D+1}(M, \mathbb{Z})$.

When $M = BG$ for group $G$, such attachments are discussed in [19], and in particular the case $M = BU(1)$ in 3+1D corresponds to the Witten effect [36].

### 3.1.3 Example: $M = T^N$

Let us illustrate the previous discussions using the example of $M = T^N$ sigma model in 2+1D, with sigma model field $\lambda = (\lambda_1, \lambda_2, \cdots, \lambda_N)$ that satisfy $\lambda_i \sim \lambda_i + 2\pi$. The sigma model can arise from spontaneously broken $U(1)^N$ symmetry. Consider the topological action

$$\int \omega^{(3)} = \kappa_{ijkl} \int \lambda_i \frac{d\lambda_j}{2\pi} \frac{d\lambda_k}{2\pi} \frac{d\lambda_l}{2\pi} , \tag{3.5}$$

where the coefficient $\kappa_{ijkl}$ is an integer that is antisymmetric with respect to permutations of the indices.

Consider the effect of the topological interaction $\omega^{(3)}$ on the following magnetic defects:

- Isometry defects, consider the subgroup $\mathrm{Isom}(M) \supset U(1)^N$ generated by $\lambda_i \to \lambda_i + \alpha_i$ for $\alpha_i \in \mathbb{R}/2\pi\mathbb{Z}$. The isometry defect $\alpha = (\alpha_1, \alpha_2, \cdots, \alpha_N)$ supported on domain wall $M_2$ changes the topological action $\omega^{(3)}$, and thus it is attached to the difference

$$e^{i\alpha_i \kappa_{ijkl} \int_{V_3} \frac{d\lambda_j}{2\pi} \frac{d\lambda_k}{2\pi} \frac{d\lambda_l}{2\pi}} , \tag{3.6}$$

where $\partial V_3 = M_2$.

- The theory has codimension-two magnetic defects that carry $\oint d\lambda_i = 2\pi m_2^{(i)}$ for integer $m_2 = (m_2^{(1)}, m_2^{(2)}, \cdots, m_2^{(N)})$. The magnetic defect supported on $M_1$ is attached to

$$e^{i\kappa_{ijkl} m_2^{(i)} \int_{V_2} \lambda_j \frac{d\lambda_k}{2\pi} \frac{d\lambda_l}{2\pi}} , \tag{3.7}$$

where $\partial V_2 = M_1$.

### 3.1.4 Example: electric charge of axion-monopole

As another example, consider $M = S^1 \times BU(1)$ in 3+1D, with the topological action

$$\int \omega^{(4)} = \frac{\kappa}{2(2\pi)^2} \int \theta F^2 = -\frac{\kappa}{2(2\pi)^2} \int d\theta a da \;, \tag{3.8}$$

where $\theta \in S^1$ is the axion, and $F = da$ is the field strength of the $U(1)$ gauge field, and $\kappa$ is an integer.

Consider the consequence of the topological action on the following magnetic defects:

- Codimension-two axion string that carries $\oint d\theta = 2\pi q_A$ for integer $q_A$. Denote the worldsheet of the axion string by $\Sigma_A$, the axion string is attached to the electric defect

$$e^{\frac{\kappa q_A}{4\pi} \int_{V_3} a da} \;, \tag{3.9}$$

  where $\partial V_3 = \Sigma_A$. Thus the axion string worldsheet has quantum Hall conductance $\kappa_H = \kappa q_A$ in units of $e^2/h$ where e is the electron charge and h is the Planck constant [37], and this implies that the axion string in the presence of magnetic monopole $\oint da = 2\pi q_m$ has electric charge

$$q_e = \kappa q_A q_m \;. \tag{3.10}$$

  The coupling between axion and magnetic monopole is also discussed in [38, 39].

- Codimension-three magnetic monopole with $\oint da = 2\pi q_m$ is attached to the electric defect

$$e^{i\frac{\kappa q_m}{2\pi} \int_{V_2} a d\theta} \;, \tag{3.11}$$

  where the boundary of $V_2$ is the worldline of the magnetic monopole.

- Codimension-two Dirac string magnetic defect with parameter $\alpha \in \mathbb{R}/2\pi\mathbb{Z}$ is attached to the theta term type electric defect

$$e^{-i\frac{\kappa \alpha}{(2\pi)^2} \int_{V_3'} d\theta da} \;, \tag{3.12}$$

  where the boundary of $V_3'$ is the magnetic defect.

For instance, if we intersect a magnetic line defect with $\oint da = 2\pi q_m'$ with the electric defect (3.11), we find the local operator

$$e^{i\kappa q_m q_m' \theta} \;. \tag{3.13}$$

We note that $\theta$ converts a monopole into a dyon with electric charge $\theta/2\pi$, and this can be regarded as an "operator-valued statistical Berry phase" that depends on the profile of the axion field $\theta$.

Similarly, the intersection of two Dirac strings with parameter $\alpha, \alpha'$ emits the electric line defect

$$e^{i\frac{\kappa \alpha \alpha'}{(2\pi)^2} \int d\theta} \;. \tag{3.14}$$

Such junctions of lower-codimensional defects producing higher-codimensional defects are analogues of the junctions of defects that generate higher-group symmetries [27]. The above correlation function is not quite well-defined, since it is not invariant under $\alpha \to \alpha + 2\pi$. Instead, we should

consider junction of Dirac strings $[\alpha'], [\alpha''], [\alpha' + \alpha'']$ where $[\cdot]$ denotes the restriction to $[0, 2\pi)$, then intersecting Dirac string $\alpha$ with the junction produces the operator

$$e^{i\alpha\kappa \frac{[\alpha']+[\alpha'']-[\alpha'+\alpha'']}{2\pi} \int \frac{d\theta}{2\pi}} . \tag{3.15}$$

In Section 8, we will discuss more detailed implications for such charge-flux attachments and the generalizations to non-Abelian gauge theories coupled to axions.

## 3.2 "Higher-group" junctions for magnetic and electric defects

When a magnetic defect intersects an electric defect, it gives a magnetic defect on the worldvolume of the electric defect, and due to the topological interaction on the electric defect, this produces additional electric defect. This gives junctions that involve defects of different dimensions, similar to higher-group symmetry.

- Intersecting codimension-$k$ magnetic defect $m_k$ with $n$-dimensional electric defect $\eta^{(n)}$ produces the electric defect

$$e^{i \int i_{m_k} \eta^{(n)}} . \tag{3.16}$$

- When the magnetic defects are attached to electric defect due to topological term $\omega^{(D)}$, braiding of magnetic defects $m_k, m'_{k'}$, i.e. intersecting the magnetic defect with the electric defect on the submanifold bounding the other magnetic defect, produce the electric defect

$$e^{i \int i_{m_k} i_{m'_{k'}} \omega^{(D)}} e^{i \int i_{m'_{k'}} i_{m_k} \omega^{(D)}} . \tag{3.17}$$

**Example: $M = BG$ for finite group $G$** In such case, the higher-group structure between the electric and magnetic defects are discussed in [19].

### 3.2.1 Example: $M = T^N$

Consider intersecting the isometry defect $\alpha' = (\alpha'_1, \alpha'_2, \cdots, \alpha'_N)$ with the electric defect (3.6) attached to the isometry defect $\alpha = (\alpha_1, \alpha_2, \cdots, \alpha_N)$, the codimension-two junction emits the electric surface defect of theta term type

$$e^{i \frac{\alpha_i \alpha'_j}{2\pi} \kappa_{ijkl} \int \frac{d\lambda_k}{2\pi} \frac{d\lambda_l}{2\pi}} . \tag{3.18}$$

Similarly, consider intersecting the isometry defects $\alpha, \alpha', \alpha''$ at codimension-three junction, at the intersection point there emits the electric line defect

$$e^{i \frac{\alpha_i \alpha'_j \alpha''_k}{(2\pi)^2} \kappa_{ijkl} \int \frac{d\lambda_l}{2\pi}} . \tag{3.19}$$

Thus such junction in the sigma model describes the analogue of two-group symmetry.

### 3.2.2 Example: $M = S^2$ in 3+1D

Consider $M = S^2$ sigma model in 3+1D. Let us consider the codimension-two junction of the $SO(3)$ isometry defect describing the non-trivial element in $\pi_1(SO(3)) = \mathbb{Z}_2$. Let us further intersect the junction with the domain wall given by the Chern-Simons term of the Berry connection on $S^2$ with level $k$, to produce codimension-three junction of domain walls. Then the intersection of the codimension-two junction with the Chern-Simons domain wall produce the theta term type electric surface defect

$$e^{k\pi i \int \lambda^* \text{vol}(S^2)} \, , \tag{3.20}$$

where $\text{vol}(S^2)$ is the unit volume form on $S^2$. Thus for odd $k$ there is extra electric surface defect, while for even $k$ the defect is trivial.

Such behavior can also be reproduced from a microscopic construction with spontaneously broken $SO(3)$ symmetry. Consider $U(1)$ gauge theory with two complex scalars transformed under $SO(3)$ flavor symmetry. If we turn on a Higgs potential preserving the $SO(3)$ symmetry, the theory flows to the $M = S^2$ sigma model. In the presence of background for the $SO(3)$ symmetry that is not an $SU(2)$ background field, as specified by $w_2^{SO(3)} \in H^2(BSO(3), \mathbb{Z}_2)$ the $U(1)$ magnetic flux is quantized by multiple of $\pi w_2^{SO(3)}$ mod $2\pi$. Then on the domain wall with $U(1)_k$ Chern-Simons term, this is the same as turning on background for $\mathbb{Z}_2$ center one-form symmetry. For odd $k$, there is no one-form symmetry, which means that the dynamical field of the Chern-Simons term depends on the bulk, and the dependence is a 2d theta term with $\theta = \pi$, which is the emitted surface operator (3.20).

## 3.3 Application: comparing energy scales

If the emitted defect from the junction is not topological, then it costs energy to nucleate or deform such defects. Such deformation can be implemented by deforming the junction that emits the defect, and thus the topological condition for the constituent defects in the junction guarantees the topological condition for the emitted defect. Suppose the support of the $i$th constituent defect is deformed slightly by $\Sigma_i$, with energy cost $\Delta_{\Sigma_i} E_i$, which causes the support of the emitted defect (we will label it by 0) to deform slightly by $\Sigma_0$ (for instance, the location where the defect is emitted might be deviated), and such deformation of the emitted defect cost energy $\Delta_{\Sigma_0} E_0$. Then the energy cost to manufacture such deformation of the emitted defect using the junction is at least $\Delta_{\Sigma_0} E_0$, and this suggests the inequality:

$$\sum_i \Delta_{\Sigma_i} E_i \geq \Delta_{\Sigma_0} E_0, \qquad \Delta_{\Sigma_I} E_I \geq 0 \, , \tag{3.21}$$

where we take the change of the energy to be non-negative for the defect to be stable. Thus if the emitted defect is not topological, $\Delta_{\Sigma_0} E_0 > 0$, then $\Delta_{\Sigma_i} E_i > 0$ for some $i$. Since we can select any defects involved in the junction as the "emitted defect", we can replace defect 0 on the right hand side with any other defect specified on the left hand side, and thus the energy cost is similar to a convex function.

On the other hand, the contrary does not have to be true: $\Delta_{\Sigma_i} E_i > 0$ for some $i$ is consistent with $\Delta_{\Sigma_0} E_0 = 0$, and the junction of non-topological defects could in principle produce topological defects.

We remark that when the defects have an renormalization group flow that make them topological and invertible at low energy, this agrees with the constraint on symmetry breaking in invertible two-group global symmetry discussed in [40]. See also *e.g.* [41, 42, 19] for examples of applying such constraints.

When the defect is magnetic, we can estimate the change in the energy cost from the change in the background configuration of the sigma model field and the kinetic term together with the interactions in the action of the nonlinear sigma model.[6]

# 4 Non-invertible magnetic defects from topological interaction

## 4.1 Non-Abelian fusion of magnetic defects

As discussed in Section 3, in the presence of topological interaction $\omega^{(D)}$, the magnetic defect of codimension $k$ labelled by $[m] \in \pi_{k-1}(M)$ is attached to electric defect $i_m \omega^{(D)}$.

As a consequence of the attaching electric defect, under a change of the coordinate patches, the electric defect contributes a boundary variation given by the "anomaly descendant", where the variation is an "anomaly in the space of coupling" corresponds to the bulk Berry phase [43, 11, 10].

**Anomaly descendant** The boundary variation of a general $n$-dimensional electric defect $\eta^{(n)}$ can be computed as follows. We consider the electric defect on $D^{k-1} \times \Sigma_{n-k+1}$ for $n - k + 1 \geq 0$, and comparing the change of coordinate patch on the image of $D^{k-1}$ under the sigma model field $\lambda : X \to M$. This can be computed in the following two ways:

- The change is given by the boundary variation.
- We glue the configuration before and after the change into $S^{k-1} \times \Sigma_{n-k+1}$, with prescribed homotopy class $S^{k-1} \to M$. This gives the reduction of $\eta^{(n)}$ in the background configuration of the homotopy class.

Comparing the two, we find that the boundary variation is given by

$$i_{m_k} \eta^{(n)} \ , \tag{4.1}$$

where $m_k : S^{k-1} \to M$ prescribes the non-trivial change of coordinate charts on $M$.

### 4.1.1 Fusion of magnetic defect with electric defect

Consider fusion of codimension-$k$ magnetic defect $m_k \in \pi_{k-1}(M)$ with electric defect. If we change the coordinate chart on $M$, the electric defect $i_m \omega^{(D)}$ attached to the magnetic defect has boundary

---

[6]We remark that since the emitted electric defect can be obtained from twisted compactification in the magnetic background, there could also be Kaluza-Klein modes that become massless due to the background field configuration. We ignore these contributions in the previous discussions.

contribution for such gauge transformation that enters the fusion channel. Denote the magnetic defect by $U_{m_k}$, and electric defect for $\eta^{(n)} \in H^n(M, U(1))$ by $\mathcal{W}_{\eta^{(n)}}$, we have

$$U_{m_k}(\Sigma_{D-k}) \times \mathcal{W}_{i_{n_\ell} i_{m_k} \omega^{(D)}}(M_{D-k-\ell+2}) = U_{m_k}(\Sigma_{D-k}) \ ,$$
$$M_{D-k-\ell+2} \in H_{D-k-\ell+2}(\Sigma_{D-k}), \quad n_\ell \in \pi_{\ell-1}(M) \ . \tag{4.2}$$

### 4.1.2  Fusion of magnetic defects

Similarly, we have

$$U_{m_k}(\Sigma_{D-k}) \times U_{m'_k}(\Sigma_{D-k})$$
$$= U_{m_k+m'_k}(\Sigma_{D-k}) \times \frac{1}{\mathcal{N}} \sum \mathcal{W}_{i_{n_\ell} i_{m_k} \omega^{(D)}}(M_{D-k-\ell+2}) \mathcal{W}_{i_{n'_{\ell'}} i_{m'_k} \omega^{(D)}}(M'_{D-k-\ell'+2}) \ , \tag{4.3}$$

where the summation is over $n_\ell \in \pi_{\ell-1}(M)$, $n'_{\ell'} \in \pi_{\ell'-1}(M)$, $M_{D-k-\ell+2} \in H_{D-k-\ell+2}(M_{D-k})$, $M_{D-k-\ell'+2} \in H_{D-k-\ell'+2}(M_{D-k})$, and $\mathcal{N}$ is an overall normalization factor to ensure the trivial fusion channel only contributes once.

### 4.1.3  Invertible magnetic defects

The magnetic defects are invertible *i.e.* Abelian, if there are no anomaly in the target space, *i.e.* the bounding electric defect belongs to the trivial class:

$$\text{Abelian magnetic defects}: \quad [i_{m_k} \omega^{(D)}] = 0 \ . \tag{4.4}$$

If $\omega^{(D)}$ has finite order, the magnetic defects form an Abelian fusion algebra that is extended by the electric defects, with the extension class given by $\Omega_{m,m'} \omega^{(D)}$ in (3.4).

### 4.1.4  Example: $M = T^N$

Let us illustrate the discussion with $M = T^N$ sigma model in 2+1D, with coordinate $\lambda = (\lambda_1, \cdots, \lambda_N)$ that obeys $\lambda_i \sim \lambda_i + 2\pi$. Consider the topological action (3.5). The codimension-two magnetic defect that carries $\oint d\lambda_i = 2\pi m_2^{(i)}$ is attached to the electric defect (3.7),

$$e^{i\kappa_{ijkl} m_2^{(i)} \int_{V_2} \lambda_j \frac{d\lambda_k}{2\pi} \frac{d\lambda_l}{2\pi}} \ . \tag{4.5}$$

The magentic defect is invariant under fusion with the electric defect

$$U_{m_2}(\gamma) \times e^{i\kappa_{ijkl} m_2^{(i)} m_2'^{(j)} \int_\gamma \lambda_k \frac{d\lambda_l}{2\pi}} = U_{m_2}(\gamma) \ , \tag{4.6}$$

for any $m'_2$, since such electric defect can be removed by changing the coordinate charts.

Fusing the magnetic defects $m_2, (-m_2)$ produces

$$U_{m_2} \times \overline{U_{m_2}} = \frac{1}{\mathcal{N}} \sum_{m'_2} e^{i\kappa_{ijkl} m_2^{(i)} m_2'^{(j)} \int \lambda_k \frac{d\lambda_l}{2\pi}} \ , \tag{4.7}$$

where $\mathcal{N}$ is a normalization factor.

### 4.1.5 Example: isometry defects become non-invertible

When the isometry defect is attached to electric defect $\rho\omega^{(D)} - \omega^{(D)}$ due to topological interaction $\omega^{(D)}$, where $\rho$ denotes the isometry, the fusion rule of the isometry defect is modified. Denote the isometry defect by $U_\rho$ supported on codimension-one submanifold $\Sigma_{D-1}$

$$U_\rho(\Sigma_{D-1}) \times U_{\rho'}(\Sigma_{D-1}) = U_{\rho\rho'}(\Sigma_{D-1})\frac{1}{\mathcal{N}}\sum \mathcal{W}_{i_{m_k}(\rho\omega^{(D)}-\omega^{(D)})}(M_{D-k})\mathcal{W}_{i_{m_{k'}'}(\rho'\omega^{(D)}-\omega^{(D)})}(M'_{D-k'}) ,$$
$$(4.8)$$

where the summation is over $[m_k] \in \pi_{k-1}(M), [m_{k'}] \in \pi_{k'-1}(M)$ that are invariant under the action of isometries $\rho, \rho'$, and $M_{D-k} \in H_{D-k}(\Sigma_{D-1})$, $M'_{D-k'} \in H_{D-k'}(\Sigma_{D-1})$. The normalization factor $\mathcal{N}$ to included to ensure the trivial fusion channel only contributes once.

**Example: non-invertible time-reversal symmetry in 3+1D $PSU(N)$ Yang-Mills theory at $\theta = \pi$**  Consider $PSU(N)$ Yang-Mills theory in 3+1D with $\omega^{(D)}$ given by the $\theta = \pi$ theta term. Across the codimension-one domain wall that generates the time-reversal symmetry, the action changes by

$$\frac{2\pi(N-1)}{2N}\int \mathcal{P}(w_2^{PSU}) ,\qquad (4.9)$$

where $w_2^{PSU}$ is the $\mathbb{Z}_N$ two-form gauge field that is the obstruction to lifting the $PSU(N)$ bundle to an $SU(N)$ bundle. Such topological term admits a topological boundary condition. This implies that the codimension-one domain wall that generates the time-reversal symmetry is not invertible. Instead, we have the fusion rule for the minimal decoration on the domain wall (denote $T$ to be the time-reversal generator)

$$U_T(\Sigma_3) \times U_T(\Sigma_3) = \frac{1}{\mathcal{N}}\sum_{\gamma \in H_2(\Sigma_3, \mathbb{Z}_N)} e^{2\pi i\frac{N-1}{2N}\int_{\Sigma_3} \mathrm{PD}(\gamma)\cup d\mathrm{PD}(\gamma)} e^{\frac{2\pi i(N-1)}{N}\int_\gamma w_2^{PSU}} ,\qquad (4.10)$$

where $\mathrm{PD}(\gamma)$ denotes the Poincaré dual of $\gamma$ on $\Sigma_3$. This reproduces the result in [22, 44],

## 4.2 Non-Abelian statistics of magnetic defects and "Gauss law"

When the magnetic defects $m_k, m_{k'}'$ can braid, braiding the magnetic defects produce the electric defect

$$\mathcal{W}_{i_{m_k}i_{m_{k'}'}\omega^{(D)}} .\qquad (4.11)$$

Since additional defects are produced, after the braiding the configuration of defects does not return to the original configuration, and thus the braiding of magnetic defects become non-Abelian.

This also mean that on the Hilbert space, if we contract one magnetic operator (suppose it is topological), this annihilates the state, since the braiding produces extra defect. This can be interpreted as a Gauss law.[7]

---

[7]An example of such "Gauss law" is discussed in [45] for $M = BU(1) \times S^1$.

### 4.2.1 Example: $M = T^N$

Let us illustrate the discussion with $M = T^N$ sigma model in 2+1D, with coordinate $\lambda = (\lambda_1, \cdots, \lambda_N)$ that obeys $\lambda_i \sim \lambda_i + 2\pi$. Consider the topological action (3.5). The codimension-two magnetic defect that carries $\oint d\lambda_i = 2\pi m_2^{(i)}$ is attached to the electric defect (3.7),

$$e^{i\kappa_{ijkl}m_2^{(i)}\int_{V_2}\lambda_j\frac{d\lambda_k}{2\pi}\frac{d\lambda_l}{2\pi}} \quad . \tag{4.12}$$

If we braid two codimension-two magnetic defects with $m_2, m_2'$, where one magnetic defect intersects the surface that bounds the other magnetic defect, the braiding produces the electric line defect

$$e^{2i\kappa_{ijkl}m_2^{(i)}m_2'^{(j)}\int\lambda_k\frac{d\lambda_l}{2\pi}} \quad . \tag{4.13}$$

## 5 Correlation functions

Let us compute the statistical correlation functions of the defects. They can be thought of as non-commutativity of operator product expansion for these defects. For instance, non-trivial braiding of two defects imply that different orderings along the direction separating the two defects in their operator product expansions give different results. The statistical correlation functions correspond to the difference given by an overall complex number. When the defects are topological, the statistical correlation function represent an anomaly of the symmetry generated by the topological defects.

The statistical correlation function between electric defects are trivial. Thus it is sufficient to consider the correlation functions involving the magnetic defects. The correlation function can be computed similar to the computation about higher-group junctions; here, we insert sufficient number of magnetic defects such that the emitted electric defect is a non-trivial phase multiplied with the identity operator. The phase is the statistical correlation function.

### 5.1 Example: generalization of multiple-loop braiding

Suppose $\pi_1(M) \neq 1$ (or we can consider non-trivial codimension-two junction of the isometry defect). Consider theory with trivial topological action $\omega^{(D)} = 0$. Consider $n$-dimensional electric defect $\eta^{(n)} \in H^n(M, U(1))$. Then there is statistical correlation functions between $n$ magnetic defects $m_1^{(1)}, \cdots, m_1^{(n)}$ and the electric defect

$$\langle U_{m_1^{(1)}} U_{m_1^{(2)}} \cdots U_{m_1^{(n)}} \mathcal{W}_{\eta^{(n)}} \rangle = e^{\int \lambda^* i_{m_1^{(1)}} i_{m_1^{(1)}} \cdots i_{m_1^{(n)}} \eta^{(n)}} \quad . \tag{5.1}$$

In 3+1D, and $n = 2$, this statistical correlation function describes the three-loop braiding [46].

### 5.2 Topological and ending conditions for defects

#### 5.2.1 Ending condition

We want to ask whether the defect can end at the energy scale at or below which we define the sigma model.

Electric defects of dimension $n$ can end if the there is no winding configuration of parameter on $S^n$. In other words, if there is no magnetic defects of codimension $(n+1)$ that can braid with it, $\pi_n(M) = 1$. Similarly, magnetic defects of codimension $k$ can end if it does not braid non-trivially with any electric defect of dimension $(k-1)$. Such magnetic defects correspond to $[m] \in \pi_{k-1}(M)$ such that $h(m)$ is trivial.

More generally, a well-defined electric defect with boundary requires the vanishing of boundary terms under field variations. The variations of the fields that are "large gauge transformations" can be characterized by homotopy group element $[m_k] \in \pi_k(M)$ for $1 \leq k \leq n$, and for $\eta^{(n)} \in H^n(M, U(1))$, the first order boundary variation is given by

$$\mathrm{vol}(S^{k-1}) \cup \left( h(m_k) \cap \eta^{(n)} \right) \ , \tag{5.2}$$

where $\mathrm{vol}(S^{k-1})$ is the unit volume form on $S^{k-1}$, which is the boundary of $k$-dimensional disk. If the above variation is trivial for all $m_k$, then the electric defect can have a well-defined boundary on general $n$-dimensional submanifolds. (If we ask whether the defect can be defined on $n$-dimensional disk, then we only need to require $h(m_n) \cap \eta^{(n)} = 0$ for all $m_n$; in other words, the electric defects braid trivially with the magnetic defects).

### 5.2.2  Topological condition

The defects described above may or may not be topological. The defects that are topological braid trivially with defects that can end, and only braid non-trivially with defects that cannot end.

### 5.2.3  Example of topological condition and ending condition

Let us illustrate the topological condition and ending condition using some examples of target space $M$.

- Example $M = S^1$. The magnetic defect of codimension two that braids with the electric line defect is not topological because the electric line defect can end. The magnetic defect cannot end, and the electric defect is topological.

- Example: $M = BG$, which is equivalent to pure gauge theory with gauge group $G$. If $G$ is a finite group, $\pi_n(BG) = \pi_{n-1}(G) = 1$ for $n \geq 2$, and thus the electric defects of dimension $n \geq 2$ have boundary condition.

## 6  Coupling nonlinear sigma models to TQFTs

Let us study coupling sigma model to topological quantum field theories. We start with a family of gapped systems that flow to the same TQFT, then we promote the parameter to be dynamical sigma model fields.

### 6.1  Defect attachment

The family of gapped systems can be characterized as follows: we vary the parameter on a $k$-dimensional cycle in spacetime, tracing out some $l$-dimensional cycle on parameter space $M$, and

such cycle in spacetime intersects a codimension-$k$ topological defect of the underlying TQFT. If the topological defect has non-trivial mutual statistics with another topological defect in the TQFT, this implies that varying the parameter on the supporting submanifold of the other defect produces a phase given by the braiding between the topological defects, and thus the other topological defect has an anomaly in the parameter space [43, 11] on the submanifold supporting the defect.

After we promote the parameters to be dynamical fields, the above properties imply that the topological defects in the TQFTs are attached to defects in the sigma model, as we will discuss below.

### 6.1.1 Magnetic defects attached to topological defects

When we promote the parameter to be dynamical, this means that the corresponding topological defect can end on the magnetic defect of codimension $(k + 1)$ described by $\pi_k(M)$. In other words, the magnetic defect is attached to a topological defect in the TQFT. The magnetic defect, which may not be topological, provides the "condensation defect" of the corresponding topological defect in the TQFT.[8]

### 6.1.2 Topological defects attached to electric defects

If the submanifolds that support topological defects in the TQFT have an anomaly in the parameter space, after we promote the parameter to be dynamical fields these topological defects are attached to a bulk electric defects of the sigma model to compensate the "gauge anomaly" on the submanifolds.

### 6.1.3 Example: $M = T^N$ sigma model coupled to $\mathbb{Z}_K$ gauge theory

To illustrate the above discussion, consider sigma model with target space $M = T^N$ coupled to the $\mathbb{Z}_K$ gauge theory in $D$-dimensional spacetime. We first start with the family of gapped systems describing $\mathbb{Z}_K$ gauge theory,

$$\eta_2 \in H^2(M, \mathbb{Z}_K), \quad \nu_{D-1} \in H^{D-1}(M, \mathbb{Z}_K) . \tag{6.1}$$

This implies that the magnetic operator of codimension two in the $\mathbb{Z}_K$ gauge theory, which is charged under $(D - 2)$-form symmetry, is attached to $e^{i \int \lambda^* \nu_{D-1}}$; similarly, the $\mathbb{Z}_K$ Wilson line is attached to $e^{i \int \lambda^* \eta_2}$.

Concretely, denote the sigma model field by $\lambda = (\lambda_1, \lambda_2, \cdots, \lambda_N)$ with $\lambda_i \sim \lambda_i + 2\pi$, then $\lambda^* \eta_2 = \alpha_{ij} \frac{d\lambda_i}{2\pi} \frac{d\lambda_j}{2\pi}$, $\lambda^* \nu_{D-1} = \beta_{i_1, i_2, \cdots, i_{D-1}} \frac{d\lambda_{i_1}}{2\pi} \frac{d\lambda_{i_2}}{2\pi} \cdots \frac{d\lambda_{i_{D-1}}}{2\pi}$ with integers $\alpha_{ij}, \beta_{i_1, \cdots, i_{D-1}} \in \mathbb{Z}_K$. The $\mathbb{Z}_K$ gauge theory can be described by $U(1)$ one-form gauge field $a$ and $U(1)$ $(D - 2)$-form gauge field $b$ with the coupling

$$\int \left( \frac{K}{2\pi} a db + a \lambda^* \nu_{D-1} + b \lambda^* \eta_2 \right) . \tag{6.2}$$

---

[8]We note that the topological defect may not have topological condensation defects; nevertheless, the magnetic defect always provides a boundary condition.

**Modifying electric defects** The equation of motion for $a, b$ are

$$db = \frac{2\pi}{K} \lambda^* \nu_{D-1}, \quad da = \frac{2\pi}{K} \lambda^* \eta_2 . \tag{6.3}$$

Thus the electric defects of the sigma model $\int \lambda^* \nu_{D-1}, \int \lambda^* \eta_2$ (we can map them to theta term type electric defects) can have boundaries given by the topological defects $\int a, \int b$ in the TQFT.

**Modifying magnetic defects** The coupling implies that the codimension-two magnetic defect that carries holonomy $\oint d\lambda_i = 2\pi m_2^{(i)}$ is attached to the defect

$$e^{im_2^{(i)} \int \left( a\beta_{i,j_1,\cdots,j_{D-2}} \frac{d\lambda_{j_1}}{2\pi} \cdots \frac{d\lambda_{j_{D-2}}}{2\pi} + b\alpha_{ij} \frac{d\lambda_j}{2\pi} \right)} . \tag{6.4}$$

The codimension-three magnetic defect given by the junction of codimension-two magnetic defects $q_i, q_j$ is attached to the topological defect $e^{m_2^{(i)} m_2^{(j)} \alpha_{ij}} \int b$. In particular, for $\eta_2 \neq 0, \nu_{D-1} = 0$ (*i.e.* $\alpha \neq 0$, $\beta = 0$), the defect $\int b$ in the TQFT now can end, and the defect $\int a$ in the TQFT that has non-trivial mutual braiding with such defect $\int b$ that admits a boundary becomes non-topological.

## 6.2 Modified non-Abelian fusion and braiding of defects

Attaching the magnetic defect to topological defect in the TQFT changes the fusion algebra of the magnetic defect by the fusion algebra obeyed by the condensation defect of the topological defect. For instance, if the corresponding condensation defect obeys the fusion algebra

$$U_\mathcal{C} \times \overline{U_\mathcal{C}} = \frac{1}{\mathcal{N}} \sum U_\gamma , \tag{6.5}$$

where $U_\gamma$ are another topological defects in the TQFT and $\mathcal{N}$ is a normalization factor, then right hand side contributes to the fusion of the magnetic defect attached to the topological defect. For instance, the fusion of magnetic defects can produce topological defect in TQFT. Similarly, braiding of the magnetic defect can produce additional topological defects, and this contributes additional non-Abelian braiding channel for the magnetic defects.[9] Similarly, some topological defects in the TQFT are attached to the electric defects in the sigma model, and this changes the fusion algebra and braiding of the topological defects.

### 6.2.1 Example: $M = T^N$ sigma model coupled to $\mathbb{Z}_K$ gauge theory

Consider the sigma model coupled to $\mathbb{Z}_K$ gauge theory with non-trivial $\eta_2$, but $\nu_{D-1} = 0$ (*i.e.* $\alpha \neq 0$, $\beta = 0$). The magnetic defect $U_{m_1}$ of codimension two that carries $\int d\lambda_i = 2\pi m_2^{(i)}$ is attached to (6.4).

**Non-Abelian fusion of magnetic defects** The magnetic defect obey the following fusion rule, which can be obtained using the method in [19]:

$$U_{m_2} \times U_{m_2'} = U_{m_2+m_2'} \frac{1}{\mathcal{N}} \sum e^{im_2^{(i)} \alpha_{ik} \int_{\gamma_k} b} e^{\frac{2\pi i}{K} m_2^{(i)} \alpha_{ik} \int_\gamma \frac{d\lambda_k}{2\pi}} e^{im_2'^{(i)} \alpha_{jk} \int_{\gamma_k'} b} e^{\frac{2\pi i}{K} m_2'^{(j)} \alpha_{jk} \int_{\gamma'} \frac{d\lambda_k}{2\pi}} , \tag{6.6}$$

where $\mathcal{N}$ is an overall normalization factor to ensure the trivial fusion channel only contributes once.

---

[9]Examples of such phenomena are observed in *e.g.* [47] in 2+1D.

**Non-Abelian braiding of magnetic defects**  The braiding of magnetic defects $m_1, m_1'$ produces the topological defect

$$e^{im_2^{(i)} m_2'^{(j)} \alpha_{ij} \int b} \, , \tag{6.7}$$

and thus the braiding of the magnetic defects does not return to the original configuration but emits an additional defect in the TQFT, this represents a Non-Abelian braiding of the magnetic defects.

# 7  Application: symmetry breaking phases in gauge theories

In this section, we will discuss examples of gauge theories with spontaneously broken continuous 0-from symmetry, and matches the symmetry and anomaly between the UV gauge theory and the IR sigma model.

## 7.1  Example: QED with $N_f$ scalars and Max$(2, D-2)$-group symmetry

Consider $U(1)$ gauge theory with $N_f$ complex scalars of charge $q$ that transform under $SU(N_f)/\mathbb{Z}_{N_f}$ symmetry, where the center of $SU(N_f)$ that transform the scalars by an $N_f$ root of unity can be identified by a gauge rotation. The spacetime dimension is denoted by $D \geq 3$.

### 7.1.1  Symmetry in the UV

As discussed in [27], the theory has two-group symmetry that combines the $\mathbb{Z}_q$ one-form symmetry and $PSU(N_f)$ 0-form symmetry. In terms of their background two-form gauge field $B_2$ and one-form gauge field, the two-group symmetry can be expressed as

$$dB_2 = \text{Bock}(w_2^f) \, , \tag{7.1}$$

where $w_2^f$ is the obstruction to lifting the $PSU(N_f)$ bundle to an $SU(N_f)$ bundle, and $\text{Bock}(w_2^f) = dw_2^f/N_f$ is the Bockstein homomorphism [48]. In terms of the defects that generate the symmetry, this means that at the codimension-three trivalent junction of $w_2^f$, there emits a codimension-two defect that generate the symmetry for $B_2$. In addition, the surface defect $\int da$ also generates $U(1)$ $(D-3)$-form symmetry.

We can obtain the theory with charge $q$ from the theory with charge one by gauging a $\mathbb{Z}_q$ $(D-3)$-form symmetry. Therefore, let us first consider the theory with $q = 1$.

**Theory with $q = 1$: mixed anomaly and $(D-2)$-group symmetry**  The theory with $q = 1$ has a mixed anomaly between $U(1)$ $(D-3)$-form symmetry generated by the magnetic flux of the $U(1)$ gauge field, and the $SU(N_f)/\mathbb{Z}_{N_f}$ flavor symmetry. To see this, we can activate a background for $PSU(N_f)$ 0-form symmetry, then the identification of transformations implies that the symmetry structure of the classical action becomes

$$\frac{U(1)_{\text{gauge}} \times SU(N_f)_{\text{global}}}{\mathbb{Z}_{N_f}} \, , \tag{7.2}$$

and thus the magnetic flux of the $U(1)$ gauge field becomes fractional. The anomaly can also be seen from the correlation function of symmetry defects. Consider the codimension-two junction

of the domain wall that generates the 0-form symmetry with non-trivial $w_2^f = n$ mod $N_f$, which is the obstruction $\mathbb{Z}_{N_f}$ two-form to lifting the $PSU(N_f)$ background gauge field to an $SU(N_f)$ background gauge field. If we intersect the magnetic flux surface defect that generates the $(D-3)$-form symmetry with element $\alpha \in \mathbb{R}/2\pi\mathbb{Z}$ with the codimension-two junction, since the magnetic flux becomes fractional $n/N_f$, there must be non-trivial correlation function

$$e^{i\alpha n/N_f} \ . \tag{7.3}$$

This implies an anomaly between the symmetries generated by these defects. The above phase is not invariant under $\alpha \to \alpha + 2\pi$ or $n \to n + N_f$. To obtain a well-defined correlation function, we need to consider codimension-three junction of the codimension-one domain wall, forms by trivalent junction of the codimension-two junctions. For the codimension-two junctions characterized by $[n], [n']$ and $[n+n']$, where $[\cdot]$ denotes restriction to $0, 1, \cdots, N_f - 1$, the phase is

$$e^{i\alpha \frac{[n]+[n']-[n+n']}{N_f}} \ , \tag{7.4}$$

where we note that $\frac{[n]+[n']-[n+n']}{N_f}$ is $\text{Bock}(w_2^f)$ evaluated at the junction (Bock denotes the Bockstein homomorphism).

In spacetime dimension $D \geq 3$, we can also consider the symmetry generated by the three-dimensional defect with Chern-Simons term on submanifold $M_3$ for the $U(1)$ gauge field, labelled by integer $\kappa$. The symmetry structure implies that in the presence of non-trivial $w_2^f$, the Chern-Simons defect depends on four dimensional bulk manifold $V_4$ with $\partial V_4 = M_3$ by

$$\pi\kappa \int_{V_4} \left( \frac{da}{2\pi} - \frac{1}{N_f} w_2^f \right)^2 = \frac{\kappa}{4\pi} \int_{M_3} ada - \frac{2\pi\kappa}{N_f} \int_{V_4} \frac{da}{2\pi} w_2^f + \frac{2\pi\kappa}{2N_f} \int_{V_4} \mathcal{P}(w_2^f) \ , \tag{7.5}$$

where $\mathcal{P}$ is the Pontryagin square operation, and $a$ is the dynamical $U(1)$ gauge field. Then this implies that intersecting the Chern-Simons defect with the codimension-two junction with $w_2^f = n$ emits the surface defect from the one-dimensional intersection

$$e^{\frac{2\pi i \kappa n}{N_f} \int \frac{da}{2\pi}} \ . \tag{7.6}$$

Such junction implies that the defects form a higher group:

- For $D = 3$, the Chern-Simons defect is spacetime-filling, and the above defects generate 0-form symmetry. This junction of defects implies that the 0-form symmetry generated by the extension of the flavor symmetry by the $U(1)$ magnetic symmetry, into the 0-form symmetry $U(N_f)/\mathbb{Z}_\kappa$.

- For $D > 3$, the defect $\int da$ generates a $U(1)$ $(D-3)$-form symmetry, and the junction describes $(D-2)$-group symmetry.

The codimension-two junction also braids with the three-dimensional Chern-Simons defect to produce the phase

$$e^{\frac{2\pi i \kappa}{2N_f} \int \mathcal{P}(w_2^f) \cup \delta(V_4)^\perp} \ , \tag{7.7}$$

where $\delta(V_4)^\perp$ is the Poincaré dual for the four-manifold $V_4$ that bounds the Chern-Simons defect. The integral in the exponent is over the entire spacetime, and it is the braiding between the self-intersection of the codimension-two junction and the three-dimensional manifold that supports the Chern-Simons defect. For instance, in $D = 4$ this is the braiding between the intersection point of the surface, and the domain wall.

**Theory with general charge $q$ from gauging $\mathbb{Z}_q$ symmetry: two-group symmetry**   If we gauge the $(D-3)$-form $\mathbb{Z}_q \subset U(1)$ symmetry, we introduce the coupling

$$\int \frac{da}{2\pi} b_{D-2} \; , \tag{7.8}$$

where $b_{D-2}$ is the dynamical $\mathbb{Z}_q$ $(D-2)$-form gauge field, and $e^{i\int b_{D-2}}$ can be interpreted as the vortex string of $\mathbb{Z}_q$ gauge theory that couples to the theory with $q = 1$, and it generates $\mathbb{Z}_q$ one-form symmetry that transforms the Wilson line $e^{i\int a}$. Let us discuss how the coupling modifies the junctions.

In the codimension-three junction of the flavor symmetry defects form by the trivalent junction of the codimension-two junctions with $w_2^f = [n], [n'], [n + n']$, the phase depends on the coefficient $\alpha$ of the defect $e^{\frac{i\alpha \int da}{2\pi}}$. After gauging the $\mathbb{Z}_q$ symmetry, the phase becomes a non-trivial operator: for $\alpha = 2\pi\ell/q$ with $\ell \in \mathbb{Z}_q$, this is the same as emitting the operator $e^{i\int b_{D-2}}$ where $b_{d-2} = (2\pi\ell/q)\delta(\Sigma)^\perp$ is the corresponding $(D-2)$-form gauge field, where $\Sigma$ is the surface supporting the magnetic flux. Since $n$ is defined modulo $N_f$, this is not quite well-defined; instead, we should consider the codimension-three junction of the domain walls where three codimension junctions with $[n], [n'], [n + n']$ meet (bracket denotes restricting the range to $0, \cdots, N_f - 1$), then the junction emits the well-defined operator

$$e^{2\pi i \frac{[n]+[n']-[n+n']}{N_f} \int b_{D-2}} \; . \tag{7.9}$$

Since $\frac{[n]+[n']-[n+n']}{N_f}$ is the definition of the Bockstein map, this reproduces the two-group relation (7.1).

The junction between the Chern-Simons defect and the junction of the flavor symmetry defect emits the operator $e^{\frac{2\pi i \kappa n}{N_f} \int da}$. This operator is in general not an integer power of the generator $e^{\frac{2\pi i}{q} \int \frac{da}{2\pi}}$ for the $\mathbb{Z}_q$ symmetry we gauged, except for the Chern-Simons defects with $\kappa \in N_f / \gcd(N_f, q)\mathbb{Z}$, and thus the junction still implies the theory has $(D-2)$-group symmetry. However, there is new anomaly given by new correlation function involving the junction.

To see the new correlation function, we note that the emitted operator $e^{\frac{2\pi i \kappa n}{N_f} \int da}$ can intersect with the codimension-two operator $e^{i\int b_{D-2}}$ to produces a phase. Thus the junction implies the new correlation function

$$e^{2\pi i \frac{\kappa}{N_f^2} \int \delta(M_{D-2})^\perp \cup \delta(V_4)^\perp \cup w_2^f} \; , \tag{7.10}$$

where $M_{D-2}$ is the support of the operator $e^{i\int b_{D-2}}$ and it si bounded by some $(D-1)$-manifold $V'_{D-1}$, and $V_4$ is the four-manifold that bounds the Chern-Simons defect. This is not quite well-defined unless $\kappa$ is a multiple of $N_f$, and thus we need to consider the codimension-three junction of

codimension-two junction:

$$e^{2\pi i \frac{\kappa}{N_f} \int \delta(V_{D-1})^{\perp} \cup \delta(V_4)^{\perp} \cup \mathrm{Bock}(w_2^f)} .$$

(7.11)

### 7.1.2 Matching symmetry in the IR sigma model

Suppose the scalar condenses, this gives sigma model with target space $M = SU(N_f)/U(N_f - 1)$. Since the symmetry in the theory with charge $q$ is related to the theory with charge one by gauging $\mathbb{Z}_q$ symmetry, it is sufficient to consider the case $q = 1$. We will show how the mixed anomaly is reproduced in the sigma model. For general $q$, the low energy theory will be sigma model coupled to $\mathbb{Z}_q$ gauge theory TQFT.

**Mixed anomaly and $(D-2)$-group symmetry in theory with $q = 1$** The magnetic defects are given by homotopy groups $\pi_{k-1}(M)$, which can be computed by the homotopy long exact sequence for $U(N_f - 1) \to SU(N_f) \to M$. The defects are matched between the UV and the IR as follows

- The equation of motion for the UV gauge field matches the UV magnetic flux with the Kähler form on $M$, and thus the UV $U(1)$ $(D-3)$-form symmetry generated by the magnetic flux matches to the $U(1)$ symmetry generated by the theta-term type electric defect of dimension two $H^2(M, \mathbb{Z})$ in the sigma model.

- The flavor symmetry is matched with the codimension-one magnetic defect for the isometry on $M$.

The codimension-one magnetic defect can form codimension-two junction, which corresponds to the Dirac string type magnetic defect for $\pi_2(M) = \mathbb{Z}$ with fraction $n/N_f$ for $n = w_2^f$. Since the generator of $\pi_2(M)$ has unit pairing with the generator of $H^2(M)$, when the theta term type electric surface defect $H^2(M)$ corresponds to $\alpha \in \mathbb{R}/2\pi\mathbb{Z}$ intersects the codimension-two the codimension-two junction of magnetic defect at a point, it produces the same correlation function as the defects in the UV:

$$e^{i\alpha n/N_f} .$$

(7.12)

As in the previous discussion, the above correlation function is not quite well-defined, but we can form well-defined correlation function by considering the codimension-three trivalent junction for the codimension-two junctions with $[n], [n'], [n + n']$, then the correlation function is given by

$$e^{i\alpha \frac{[n]+[n']-[n+n']}{N_f}} .$$

(7.13)

**General $q$: coupling sigma model to TQFT** For the case of general $q$, the IR is sigma model coupled to $\mathbb{Z}_q$ gauge theory. The $\mathbb{Z}_q$ one-form symmetry is generated by the vortex string of the $\mathbb{Z}_q$ gauge theory.

The sigma model couples to the TQFT by attaching the magnetic monopole of codimension three $\pi_2(M) = \mathbb{Z}$ to the vortex string of the $\mathbb{Z}_q$ gauge theory. (To see this, we note that the coupling $\int \frac{da}{2\pi} b_{D-2}$ in the UV implies the following: the codimension-three monopoles that braid with $\int da$ are attached to the defect $\int b_{D-2}$.) The $\mathbb{Z}_q$ Wilson line is no longer topological. As a consequence of

the attachment, the codimension-three junction of the codimension-one isometry defect that hosts the codimension-three magnetic defect emits the vortex string of the $\mathbb{Z}_q$ gauge theory, and thus the defects form the junction of a two-group symmetry.

To see the $(D-2)$-group junction, we note that the Chern-Simons defect is the Chern-Simons term of the Berry connection on $M$ whose field strength is the generator of $H^2(M, \mathbb{Z})$. Then using $i_m$ on the Chern-Simons term of the Berry connection for $m$ given by $w_2^f = n$ produces the electric surface defect of theta term type with theta angle $\frac{\kappa n}{N_f}$, as the junction in the UV.

## 7.2 Example: QCD$_4$ with higher-group symmetry and chiral symmetry breaking

As another example of matching symmetry between the UV and the IR, let us consider non-Abelian $Spin(N_c)$ gauge theory with $N_f$ left handed Weyl fermions in the vector representation that transform under chiral symmetry. Let us focus on the case of even $N_c, N_f$, and $N_c = 2$ mod 4, while $N_f = 0$ mod 4. The fermions transform under $SU(N_f)/\mathbb{Z}_2$ chiral symmetry, where the $\mathbb{Z}_2$ symmetry that slips the sign of all fermions is identified with a gauge rotation in $SO(N_c)$.

For small enough $N_f$, the chiral symmetry is believed to be spontaneously broken, giving rise to a nonlinear sigma model with target space $SU(N_f)/SO(N_f)$ [49]. We would like to explore evidence for such symmetry breaking by matching the symmetry between the UV and the IR. This is partially done in [50].

### 7.2.1 Symmetry in the UV

In addition to the chiral symmetry, the theory has $\mathbb{Z}_2$ charge conjugation 0-form symmetry that acts on the gauge field $a_{ij}$ and fermion $\psi_{iI}$, where $i, j$ are color indices and $I$ is the flavor index, as

$$\begin{aligned}
\psi_{1I} &\to -\psi_{1I}, \quad \psi_{iI} \to \psi_{iI} \quad \text{for } i \neq 1 \\
a_{1i} &\to -a_{1i}, \quad a_{ij} \to a_{ij} \quad \text{for } i, j \neq 1 \ .
\end{aligned} \tag{7.14}$$

The $\mathbb{Z}_2$ charge conjugation symmetry transforms linearly the baryon operator $\epsilon^{i_1, \cdots, i_{N_c}} \psi_{i_1 I_1} \cdots \psi_{i_{N_c} I_{N_c}}$.

The theory also has $\mathbb{Z}_2$ one-form symmetry $Z(Spin(N_c)) = \mathbb{Z}_4$ broken to $\mathbb{Z}_2$ by the presence of the fermion fields in the vector representation. The one-form symmetry transforms linearly on the Wilson line operators in the spinorial representations of $Spin(N_c)$.

As discussed in [51], the 0-form and one-form symmetries form a two-group symmetry. If we turn on background gauge field $B_2$ for the one-form symmetry, and background gauge field $A$ for the $\mathbb{Z}_2$ charge conjugation symmetry, and background $A^f$ for the chiral symmetry $SU(N_f)/\mathbb{Z}_2$, the background gauge fields satisfy the relation

$$dB_2 = w_2^f \cup A + \text{Bock}(w_2^f) \ , \tag{7.15}$$

where $w_2^f$ is the background $\mathbb{Z}_2$-valued two-form that describes the obstruction to lifting the background $SU(N_f)/\mathbb{Z}_2$ gauge field $A'$ to an $SU(N_f)$ gauge field. The background gauge field with non-trivial $w_2^f$ corresponds to the codimension-two junction of domain wall that generates the

chiral symmetry with elements tracing a non-contractible loop $\pi_1(SU(N_f)/\mathbb{Z}_2) = \mathbb{Z}_2$. In the above formula, $\mathrm{Bock}(w_2^f) = dw_2^f/2 \bmod 2$ is the Bockstein homomorphism.

Let us describe (7.15) in terms of the defects that generate the symmetry. The first term on the right hand side of (7.15) means that at the one-dimensional intersection between the the domain wall defect generating the 0-form symmetry corresponds to $A$, and the codimension-two junction of the flavor symmetry given by non-trivial $w_2^f$, there emits the surface defect that generates the one-form symmetry corresponds to $B_2$. The second term on the right hand side of (7.15) means that at the trivalent junction of the codimension-two junctions with $w_2^f = [n], [n'], [n+n']$ where $[\cdot]$ denotes the restriction to $0, 1$, there emits the surface defect that generates the one-form symmetry corresponds to $B_2$. It can also be interpreted as the codimension-two junction with $w_2^f = [n]$ is attached to fractional $[n]/2$ of the surface defect that generates the one-form symmetry.

### 7.2.2 Matching symmetry in the IR sigma model

Let us explore how the symmetries in the microscopic gauge theory matches in the low energy sigma model with $M = SU(N_f)/SO(N_f)$. The defects match between the UV and the IR as follows:

- The UV chiral symmetry corresponds in the IR to the isometry on the target space $M$.

  The chiral symmetry that transforms the left-handed fermions is anomalous, and this is matched by the presence of Wess-Zumino term for the sigma model.

- The UV $\mathbb{Z}_2$ charge conjugation 0-form symmetry with background gauge field $A$ corresponds in the IR to the 0-form symmetry generated by the electric domain wall defect $H^3(M, U(1)) = \mathbb{Z}_2$, which can be understood as $\mathbb{Z}_2$ valued Wess-Zumino term of $M$.

  If we regard the $M = SU(N_f)/SO(N_f)$ sigma model as gauging the $SO(N_f)$ symmetry in $SU(N_f)$ sigma model, then the domain wall defect describes the Chern-Simons term of $SO(N_f)$ gauge field that is even or odd.

- The UV baryon operators that transform under charge conjugation 0-form symmetry corresponds to the codimension-four magnetic defect $\pi_3(M) = \mathbb{Z}_2$.

- The UV Wilson lines transformed under the one-form symmetry corresponds in the IR to the codimension-three magentic defect $\pi_2(M) = \mathbb{Z}_2$.

- The surface defect that generates the one-form symmetry in the UV corresponds in the IR to the electric surface defect $H^2(M, U(1)) = \mathbb{Z}_2$.

  If we regard the $M = SU(N_f)/SO(N_f)$ sigma model as gauging the $SO(N_f)$ symmetry in $SU(N_f)$ sigma model, then the surface defect measures the $\pi_1(SO(N_f)) = \mathbb{Z}_2$ magnetic flux of the $SO(N_f)$ gauge field in this presentation. Let us denote the surface by $e^{\pi i \int w_2'}$ for dynamical $\mathbb{Z}_2$ two-form $w_2'$ for the $SO(N_f)$ gauge field.

The junction describing the two-group symmetry (7.15) in the UV is reproduced in the IR as follows:

- Consider the first term on the right hand side of (7.15) in the IR. When the codimension-two junction of the codimension-one magnetic defect intersects the electric domain wall defect, it creates a magnetic line defect on the domain wall.

If we regard the $M = SU(N_f)/SO(N_f)$ sigma model as gauging $SO(N_f)$ symmetry in $SU(N_f)$ sigma model, then the domain wall is an odd Chern-Simons term of $SO(N_f)$ gauge field. For such Chern-Simons term, the line defect in the representations that transform non-trivially under the $\mathbb{Z}_2$ center of $SO(N_f)$, is attached to the surface $e^{\pi i \int w_2'}$. In other words, the intersection emits the electric surface defect. This matches with the contribution to the left hand side of (7.15).

- Consider the second term on the right hand side of (7.15) in the IR. The matching of this contribution to the left hand side of (7.15) in the IR is discussed in [50], and we will not give the details. The codimension-three junction of the isometry defect corresponds to $\pi_2(M) = \mathbb{Z}_2$. The Wess-Zumino term implies that the junction is attached to a codimension-two electric defect, which is the electric surface defect $H^2(M, U(1))$.

# 8 Examples: defects in axion gauge theory in 3+1D

Let us consider the class of example of axions coupled to gauge field in 3+1D, such as $SU(3)$ gauge theory. We remark that such theories are originally proposed as a dynamical solution for the strong CP problem with small CP violation theta angle in quantum chromodynamics [52].

For Abelian gauge groups, the defects in the axion $U(1)$ gauge theory are discussed in *e.g.* [13, 45]. Examples with non-Abelian gauge groups are also discussed in [13, 10].

To address the problem in a more universal way with various gauge groups, we will study the defects using the bulk approach: we study the theory as the boundary of a bulk two-form gauge theory, where the gauge fields correspond to the gauge field for the one-form symmetry on the boundary. When the theta angle is a classical field, axion gauge theory can have an anomaly in the space of coupling involving the one-form symmetry, and the bulk describes such an anomaly. We will infer the property of the defects in the axion gauge theory from the defects in the bulk theory.

## 8.1 Bulk 4+1D two-form gauge theory

Let us consider two-form $\mathbb{Z}_N$ gauge theory in 4+1D with $S^1$ scalar $\theta$, such that varying $\theta$ over spacetime produces the response

$$\frac{p}{2N} \int d\theta \mathcal{P}(b) \ , \tag{8.1}$$

where $b$ is the $\mathbb{Z}_N$ two-form gauge field. Such theory is discussed in [10].

Let us study the 3+1D boundary, with the following boundary condition

$$b| = \frac{N}{\ell} b' + B_e \bmod N \ , \tag{8.2}$$

for dynamical $b'$ and classical $B_e$, where $\ell$ is a divisor of $N$. If we turn off the classical field $B_e$, then the boundary condition can be written as $b| = 0 \bmod N/\ell$. We can choose the corresponding polarization on the boundary to obtain a well-defined boundary theory [42]. In the resulting 3+1D boundary theory, $\theta$ is the axion.

## 8.2 Boundary topological defects from the bulk perspective

Let us investigate the implication of the interaction (8.1) on the boundary. Substituting the decomposition of $b$ in terms of $b', B_e$ (which we can always do for $\mathbb{Z}_N$ variable $b$), we can rewrite the topological term as follows,

$$\frac{p}{2N} \int d\theta \left( \frac{N^2}{\ell^2} \mathcal{P}(b') + \mathcal{P}(B_e) + \frac{2N}{\ell} b' \cup B_e \right) . \tag{8.3}$$

- **Non-invertible domain wall.** The first term implies that the domain wall on the boundary that generates $\theta \to \theta + 2\pi$ is attached to

$$e^{2\pi i \frac{Np}{2\ell^2} \int \mathcal{P}(b')} . \tag{8.4}$$

  If this term is non-trivial, *i.e.* $Np \neq 0 \mod \ell^2$, then the domain wall becomes non-invertible. Moreover, there are $\mathbb{Z}_\ell$ line operators on the domain wall that attach to $\int b'$ such that they have spin $pN/\ell \mod 1$. In such case, for the minimal decoration on the domain wall, the domain wall obeys the fusion rule

$$U_D(W) \times \overline{U_D(W)} = \frac{1}{\mathcal{N}} \sum_{\Sigma \in H_2(W, \mathbb{Z}_\ell)} e^{i\pi Np/\ell^2 \int \mathrm{PD}(\Sigma) \cup d\mathrm{PD}(\Sigma)/\ell} e^{\frac{2\pi i (Np/\ell^2)}{\ell} \int_\Sigma b'} , \tag{8.5}$$

  where $\mathcal{N}$ is an overall normalization factor, and the domain wall is supported on submanifold $W$.

- **Higher-group junction.** The second term implies that at the self-intersection of the surface defect described by $B_e$ with element $\alpha \in \frac{2\pi}{N/\ell} \mathbb{Z}$ such that $\int \mathcal{P}(B_e) = n_\#$, there emits the line defect

$$e^{i \frac{n_\# \alpha^2 Np}{2(2\pi)^2 \ell^2} \int d\theta} . \tag{8.6}$$

  The property that the intersection of lower-codimensional defects can produce higher-codimensional defect is a property of higher-group structure.

- **Three-loop braiding correlation function.** The emitted operator $\int d\theta$ can then braid with the axion string. This implies that there is non-trivial correlation function between the axion string $S_{q_A}$ that carries $\oint d\theta = 2\pi q_A$ and the surface defect corresponds to $B_E$ with element $\alpha \in \frac{2\pi}{N/\ell} \mathbb{Z}$:

$$\langle S_{q_A}(\Sigma_A) U_{E,\alpha}(\Sigma_E) \rangle = e^{i \frac{q_A \alpha^2 Np}{4\pi \ell^2} \int_{4d} \delta(V_A)^\perp \cup \delta(V_E)^\perp \cup \delta(\Sigma_E)^\perp} , \tag{8.7}$$

  where $V_A$ is the volume that bounds the worldsheet $\Sigma_A$ of the axion string, $\Sigma_E$ is the support of the surface defect corresponding to $B_E$, $V_E$ is the volume that bounds $\Sigma_E$, and the integral in the exponent is the triple linking number of the surface $\Sigma_A, \Sigma_E$

$$\mathrm{Tlk}(\Sigma_A, \Sigma_E, \Sigma_E) = \int_{4d} \delta(\Sigma_A)^\perp \cup \delta(V_E)^\perp \cup \delta(\Sigma_E)^\perp . \tag{8.8}$$

- **Fermionic string.** We remark that this implies that in terms of $\alpha = \frac{2\pi}{N/\ell} q_\alpha$, for $(p/N) q_A q_\alpha$ equals an odd integer, the composition $S_{q_A}(\Sigma) U_{E,\alpha}(\Sigma)$ is a surface defect that creates fermionic loop excitation [53], since the correlation function becomes (denote $V$ to be the volume that bounds the surface $\Sigma$)[10]

$$(-1)^{\int_{4d} \delta(V)^\perp \cup \delta(V)^\perp \cup \delta(\Sigma)^\perp} , \tag{8.9}$$

which can be written as a bulk term $e^{iS_{\text{bulk}}}$ using $d\delta(V)^\perp = \delta(\Sigma)^\perp$, with

$$
\begin{aligned}
S_{\text{bulk}} &= \pi \int_{5d} \left( \left( \delta(V)^\perp \cup \delta(\Sigma)^\perp + \delta(\Sigma)^\perp \cup \delta(V)^\perp \right) \cup \delta(\Sigma)^\perp \right) \\
&= \pi \int_{5d} \left( \delta(\Sigma)^\perp \cup_1 \delta(\Sigma)^\perp \right) \cup \delta(\Sigma)^\perp \\
&= \pi \int_{5d} \delta(\Sigma)^\perp \cup w_3(TM) \bmod 2\pi\mathbb{Z} ,
\end{aligned}
\tag{8.10}
$$

where $w_3(TM)$ is the third Stiefel-Whitney class of the tangent bundle, and the last equality uses $\delta(\Sigma)^\perp \cup_1 \delta(\Sigma)^\perp = Sq^1 \delta(\Sigma)^\perp$ and Appendix C of [42]. This implies that the composite surface defect $S_{q_A}(\Sigma) U_{E,\alpha}(\Sigma)$ creates fermionic loop excitations.[11]

- **Non-invertible surface defect.** The third term implies that the surface defect corresponds to $B_e$ with the element $\alpha \in \frac{2\pi}{N/\ell}\mathbb{Z}$ lives on the boundary of the volume operator

$$e^{i\alpha \frac{Np}{\ell^2} \int_{V_E} \frac{d\theta}{2\pi} b'} . \tag{8.11}$$

If $Np \neq 0 \bmod \ell$, then this implies that the defect corresponds to $B_e$ is not invertible. For the minimal decoration on the defect, the electric defect obeys the fusion rule: (take $\Sigma$ to be connected)

$$U_{E,\alpha}(\Sigma) \times \overline{U_{E,\alpha}}(\Sigma) = \frac{1}{\mathcal{N}} \sum_{n\in\mathbb{Z}} \sum_{\gamma\in H_1(\Sigma,\mathbb{Z}_\ell)} e^{i\alpha \frac{nNp}{\ell^2} \int_\Sigma b'} e^{i\alpha \frac{Np}{\ell^2} \int_\gamma \frac{d\theta}{2\pi}} , \tag{8.12}$$

where $\mathcal{N}$ is an overall normalization factor.

Moreover, the junctions of various defects can be studied using the method in [19].

- **Higher-group like junction.** Consider intersecting the domain wall that generates the shift of $\theta$ with a magnetic line defect (for instance, in gauge theory it can be realized by 't Hooft lines) that carries $\oint b' = q_m \in \mathbb{Z}_\ell$. Then at one side of the domain wall, the 't Hooft line is attached to the surface operator

$$e^{2\pi i q_m \frac{Np}{\ell^2} \int_{\Sigma_M} b'} . \tag{8.13}$$

This implies that the magnetic line defect carries "electric charge" specified by the surface, similar to the Witten effect [36].

---

[10]The author thanks Anton Kapustin for a discussion on related topics about fermionic strings and axion strings in the context of Higgsing a $U(1)$ gauge theory to $\mathbb{Z}_2$ gauge theory in 3+1D.

[11]Examples of lattice model with such fermionic loop excitations on the boundary are discussed in [53, 54], where the boundary also has a fermionic particle that has $\pi$ mutual statistics with the fermionic loop, and this implies the boundary has a gravitational anomaly.

- **Identification of the electric charge on line defects.** As a consequence, for a "dyonic" line with magnetic charge $q_m$, the electric charge (the line defect with electric charge is defined as the boundary of $e^{\frac{2\pi i q_e}{\ell} \int b'}$, for reason that we will clarify when we discuss examples of boundary axion gauge theory) can be changed by $q_e \sim q_e + \frac{Np}{\ell} q_m$.

- **Correlation function: self-intersection of surfaces braid with domain wall.**
  The emitted surface operator $\int b'$ can also braid with another magnetic line defect. This implies that the magnetic line defects and the domain wall have non-trivial correlation function. Denote the surface that bounds the support $\gamma$ of the magnetic line defect by $\Sigma_M$, and the 4-dimensional submanifold that bounds the domain wall $W$ by $R$, then the correlation function is

$$\langle U_D(W) W_{(0,q_m)}(\gamma) \rangle = e^{2\pi i q_m^2 \frac{Np}{2\ell^2} \int_{4d} \delta(R)^\perp \cup \mathcal{P}(\delta(\Sigma_M)^\perp)} , \tag{8.14}$$

  where $\mathcal{P}$ is the Pontryagin square operation. The integral in the exponent is the same as braiding the self intersection point of the surface $\Sigma_M$ with the domain wall.

- **Non-Abelian braiding of surfaces.** If we braid the axion string that carries $\oint d\theta = 2\pi q_A$ with the surface defect corresponds to $B_e$ labelled by element $\alpha \in \frac{2\pi}{N/\ell} \mathbb{Z}$, the braiding produces the surface defect

$$e^{i\alpha q_A \frac{Np}{\ell^2} \int b'} . \tag{8.15}$$

  Thus the strings obey non-Abelian mutual statistics: after we braid the two loops, it does not return to the original configuration, but there is additional defect. This also implies that on Hilbert space if we contract the axion string or contract the surface defect on the state created by other defect, we annihilate the state. This is some kind of non-invertible Gauss law.

- **Three-loop braiding correlation function.** Such surface defect can furthermore braid with the magnetic line defect that carries holonomy $\oint b'$. This means that the axion string that carries $\oint d\theta = 2\pi n_A$, the surface defect corresponds to $B_e$ with element $\alpha \in \frac{2\pi}{N/\ell} \mathbb{Z}$, and the magnetic line defect that carries $\oint b' = q_m \in \mathbb{Z}_\ell$, have non-trivial correlation function. Let us separate out the contribution in (8.7) by choosing the surface $\Sigma_E$ that does not has self braiding $\delta(V_E) \cup \delta(\Sigma_E) = 0$ for volume $V_E$ that bounds the surface $\Sigma_E$. The correlation function is:

$$\langle U_{E,\alpha}(\Sigma_E) S_{q_A}(\Sigma_A) W_{(0,q_m)}(\gamma) \rangle = e^{i\alpha q_A q_m \frac{Np}{\ell^2} \int_{4d} \delta(V_E)^\perp \cup \delta(V_A)^\perp \cup \delta(\Sigma_M)^\perp} , \tag{8.16}$$

  where $\Sigma_M$ is the surface that bounds the magnetic line defect supported on $\gamma$, $V_A$ is the volume that bounds the worldsheet $\Sigma_A$ of axion string $S_{q_A}$ that carries $\oint d\theta = 2\pi q_A$, and $V_E$ is the volume that bounds the surface defect supported on $\Sigma_E$ corresponding to $B_E$. The integral in the exponent is the triple linking number

$$\int_{4d} \delta(V_E)^\perp \cup \delta(V_A)^\perp \cup \delta(\Sigma_M)^\perp = \text{Tlk}(\Sigma_E, \Sigma_A, \Sigma_M) . \tag{8.17}$$

  Equivalently, it is the three-loop braiding number of the surfaces $\Sigma_A, \Sigma_E, \Sigma_M$. To explain where the integral comes from, we notice that braiding of the surface defect for $B_E$ and the axion string is the same as intersection of $\Sigma_A$ with $V_E$, which is a curve, and from this curve

emits the $\int b'$ surface operator supported on $\Sigma'$ whose boundary is the intersection $\Sigma_A \cap V_E$. Thus this surface can be taken to be the intersection $\Sigma' = V_A \cap V_E$. This emitted surface operator then braids with the magnetic line defect. The braiding number is given by the intersection number of the support of $\int b'$ with the surface $\Sigma$, in other words, the exponent in the correlation function has coefficient

$$\#(V_E, V_A, \Sigma_M) = \int_{4d} \delta(V_E)^\perp \cup \delta(V_A)^\perp \cup \delta(\Sigma_M)^\perp \ , \tag{8.18}$$

and this describes the three loop braiding between the axion string excitation, the magnetic loop excitation for $B_E$, and the magnetic line defect.

When the surface $\Sigma_E$ self braids, the correlation function has additional contribution (8.7) between the axion string and the surface defect that corresponds to $B_E$.

- **Non-Abelian braiding.** If we braid the magnetic line that carries $\oint b' = q_m \in \mathbb{Z}_\ell$ with the surface defect that corresponds to $B_e$ labelled by element $\alpha \in \frac{2\pi}{N/\ell}\mathbb{Z}$, it produces the line defect

$$e^{i\alpha q_m \frac{Np}{\ell^2} \int \frac{d\theta}{2\pi}} \ . \tag{8.19}$$

Thus the line and the surface defects have non-Abelian mutual statistics.

- **Three-loop braiding correlation function.** The emitted line defect $\oint d\theta$ can braid with the axion string, and this reproduces the three-loop braiding correlation function (8.16).

## 8.3 Example: axion $SU(N)/\mathbb{Z}_\ell$ gauge theory in 3+1D boundary

Consider $SU(N)/\mathbb{Z}_\ell$ gauge theory in 3+1D with axion, where $\ell$ is a divisor of $N$. We note that the Standard Model with axion is an example of such theory, where the gauge group can be $(SU(3)_C \times SU(2)_W \times U(1)_Y)/\mathbb{Z}_\ell$ with different possible $\ell = 1, 2, 3, 6$ [55].

We can apply the discussion in Section 8.2, with the identification

- The same $N, \ell$ in Section 8.2.
- The discrete parameter $p = N - 1$ in (8.1).
- $b'$ is given by the $\mathbb{Z}_\ell$ valued degree two-class that is the obstruction to lifting the $SU(N)/\mathbb{Z}_\ell$ bundle to an $SU(N)$ bundle.

**Electric charge** Since $b'$ is the gauge field for gauging the one-form symmetry in $SU(N)$ gauge theory, the Wilson lines in the representation whose Young tableaux have number of boxes $q_e$ not equal a multiple of $\ell$ is attached to the surface operator

$$e^{\frac{2\pi i q_e}{\ell} \int b'} \ . \tag{8.20}$$

The number of boxes $q_e$ is identified with the electric charge $q_e$ in Section 8.2.

The surface operator $\oint b'$ is the surface operator that generates the $\mathbb{Z}_\ell$ magnetic one-form symmetry in the theory.

- The surface defect corresponds to $B_E$ is the magnetic surface defect [56, 57, 58] with holonomy in the center $\mathbb{Z}_{N/\ell}$.

**Non-invertible magnetic surface defect with center-valued holonomy.** As discussed in Section 8.2, such magnetic surface defect becomes non-invertible due to the axion-instanton action, even though it carries holonomy that takes value in the center. Similar phenomenon in finite group gauge theory is discussed in [19], where the magnetic defect with center-valued holonomy becomes non-invertible due to the Dijkgraaf-Witten topological interaction of the gauge field.

## 8.4 Example: axion $SO(M)$ gauge theory in 3+1D boundary

Let us consider $SO(M)$ gauge theory with axion in 3+1D. The theory can be obtained from $Spin(M)$ gauge theory by gauging $\mathbb{Z}_2$ center one-form symmetry, where the total one-form symmetry is $\mathbb{Z}_2, \mathbb{Z}_4, \mathbb{Z}_2 \times \mathbb{Z}_2$ for odd $M$, $M = 2$ mod 4 and $M = 0$ mod 4. The corresponding 4+1D two-form gauge theory has $\mathbb{Z}_2, \mathbb{Z}_4$ and $\mathbb{Z}_2 \times \mathbb{Z}_2$ two-form gauge field.

### 8.4.1 Odd $M$

Since $Spin(M)/\mathbb{Z}_2 = SO(M)$, the theory can be obtained from $Spin(M)$ theory by gauging the center one-form symmetry with dynamical two-form gauge field, and thus the theory can be a boundary for the axion coupled to two-form $\mathbb{Z}_2$ gauge theory. All the discussions in Section 8.2 apply to the theory:

- $N = 2, \ell = 2$.

- The discrete parameter $p = 2$ in (8.1).

- The gauge field $b'$ is identified with the $\mathbb{Z}_2$ valued Stiefel-Whitney class of the $SO(M)$ bundle.

- There is no non-trivial magnetic surface defect that carries center-valued holonomy, since the center of $SO(M)$ is trivial for odd $M$. Correspondingly, since $N/\ell = 1$, $B_E$ can be absorbed into $b'$, and there is no non-trivial surface defect corresponds to $B_E$.

### 8.4.2 Even $M$, $M = 2$ mod 4

In such case, $Spin(M)$ has $\mathbb{Z}_4$ center one-form symmetry, and $SO(M) = Spin(M)/\mathbb{Z}_2$. The bulk theory is the same as Section 8.2, and thus we can directly apply the results there with the following dictionary:

- $N = 4, \ell = 2$.

- The discrete parameter is $p = M/2$ in (8.1).

- The gauge field $b'$ is the $\mathbb{Z}_2$ valued second Stiefel-Whitney class of the $SO(M)$ bundle, which is the obstruction to lifting the bundle to an $Spin(M)$ bundle.

- The surface defect corresponds to $B_E$ is the magnetic surface defect [58] that carries $\mathbb{Z}_2 = Z(SO(M))$ center valued holonomy.

### 8.4.3 Even $M$, $M = 0$ mod 4: bulk $\mathbb{Z}_2 \times \mathbb{Z}_2$ two-form gauge theory

**Bulk perspective**  In this case, the bulk theory is a $\mathbb{Z}_2 \times \mathbb{Z}_2$ two-form gauge theory, and let us denote the $\mathbb{Z}_2$ two-form gauge field by $b, b'$. The bulk has parameter $\theta$, with the response for spacetime-dependent $\theta$

$$\int d\theta \left( \frac{1}{2}\mathcal{P}(b') + \frac{M}{16}\mathcal{P}(b) + \frac{1}{2}b' \cup b \right) . \tag{8.21}$$

The discussion is similar to Section 8.2. Let us consider the boundary condition with dynamical $b'|$ and fixed $b| = B_e$ with background $B_e$.

Then the boundary theory has the following properties

- **Two-group like junction.** The first term of (8.21) implies that the domain wall that shifts $\theta \to \theta + 2\pi$ is attached to

$$e^{\pi i \int \mathcal{P}(b')} = e^{\pi i \int b' \cup b'} . \tag{8.22}$$

  Using the Wu formula for $b' \cup b'$, we find that the domain wall operator depends on the spin structure: if we change the local spin structure by a $\mathbb{Z}_2$ background gauge field $A$, the domain wall is decorated with

$$e^{\pi i \int b' \cup A} . \tag{8.23}$$

  We can implement the change by a domain wall that generates $\mathbb{Z}_2^f$. Then this means that at the intersection of the domain wall that shifts $\theta$ and the $\mathbb{Z}_2^f$ domain wall, there emits the surface defect

$$e^{\pi i \oint b'} . \tag{8.24}$$

  Thus we have a two-group like junction.

- **Correlation of magnetic line defect and domain wall.**
  The emitted surface operator can further braid with magnetic line defect that carries holonomy $\int b' = q'_m \in \{0, 1\}$. Denote the support of the domain wall that shifts $\theta$ by $D$ that is bounded by 4-dimensional region $R$, and the support of the magnetic line defect by $\gamma$ that is bounded by surface $\Sigma$. The braiding number of $\int b'$ and the line defect is given by the intersection of the surface that supports $\int b'$ and the surface $\Sigma$. Thus the correlation function of the domain wall $U_D$ that shifts $\theta$ and the magnetic line defect $\mathcal{W}_{(0,q_m)}$ is

$$\langle U_D(D)\mathcal{W}_{(0,q'_m)} \rangle = e^{\pi i \int \delta(R)^\perp \cup \delta(\Sigma)^\perp \cup w_2} . \tag{8.25}$$

- **Three-group like junction.** The second term in (8.21) implies that at the self intersection of the magnetic surface defect that corresponds to $B_E$, there emits the line operator

$$e^{i \frac{M n_\#}{16} \int d\theta} , \tag{8.26}$$

  where $n_\#$ is the self-intersection number. This is an analogue of a three-group junction.

- **Correlation function of magnetic surface defect and axion string.**
  The emitted line operator can braid with the axion string that carries $\oint d\theta = 2\pi q_A$ for integer $q_A$. Denote the axion string by $S_{q_A}$, which is supported on worldsheet $\Sigma_A$ that is bounded by

volume $V_A$, and denote the magnetic surface defect labelled by element $\alpha \in \{0, \pi\}$ by $U_{E,\alpha}$, supported on surface $\Sigma_E$ labelled by element that is bounded by volume $V_E$, we have the correlation function

$$\langle S_{q_A}(\Sigma_A) U_E(\Sigma_{E,\alpha}) \rangle = e^{i \frac{M q_A \alpha^2}{8\pi} \int \delta(V_A)^\perp \cup \delta(V_E)^\perp \cup \delta(\Sigma_E)^\perp} \ , \tag{8.27}$$

where the integral in the exponent is the triple linking number $\mathrm{Tlk}(\Sigma_A, \Sigma_E, \Sigma_E)$.

- **Fermionic string.** The statistical correlation function implies that For $M = 8 \bmod 16$, the composite surface defect of the axion string and the magnetic surface defect, $U_f(\Sigma) \equiv U_{q_A=1}(\Sigma) U_{E,\alpha=\pi}$ is a fermionic string.

- **Non-invertible magnetic surface defect.** The third term in (8.21) implies that the magnetic surface defect corresponds to $B_E$ labelled by element $\alpha \in \{0, \pi\}$ is attached to the volume operator

$$e^{i \frac{\alpha}{2\pi} \int_{V_E} d\theta \cup b'} \ , \tag{8.28}$$

where $V_E$ bounds the surface $\Sigma_E$ that supports the magnetic surface defect. This implies that the magnetic surface defect is non-invertible: using the inflow argument in [19], we find that for the minimal decoration on the surface to cancel the bulk dependence, we have the fusion algebra

$$U_{E,\alpha}(\Sigma) \times \overline{U_{E,\alpha}(\Sigma)} = \frac{1}{\mathcal{N}} \sum_n \sum_{\gamma \in H_1(\Sigma, \mathbb{Z}_2)} e^{i\alpha n \int_\Sigma b'} e^{i \frac{\alpha}{2\pi} \int_\gamma d\theta} \ , \tag{8.29}$$

where $\mathcal{N}$ is an overall normalization factor.

- **Identification of charges on line operator.** Shifting the theta angle by $2\pi$ shifts the set of line operators, and since the theta angle is dynamical, this gives an identification on the line operators: $(q_e, q_m) \sim (q_e + q_m, q_m)$, where $q_e, q_m = 0, 1$ denote the eigenvalues with respect to braiding the line operator with the magnetic surface defect for $B_E$ and the surface defect $e^{\pi i \int b'}$.

- **Non-Abelian braiding of magnetic line defect and magnetic surface defect.**

  If we braid a magnetic line defect that carries $\int b' = q_m \in \{0, 1\}$ with the magnetic surface defect labelled by element $\alpha$, it does not return to the original configuration, but instead there is extra line defect

$$e^{i \frac{q_m \alpha}{2\pi} \int d\theta} \ . \tag{8.30}$$

  Thus the braiding of the magnetic line defect and the magnetic surface defect becomes non-Abelian.

  Similarly, if we braid the magnetic surface operator with axion string that carries $\oint d\theta = 2\pi q_A$, *i.e.* intersecting the volume that bounds the magnetic surface operator, the braiding emits the surface operator

$$e^{i \alpha q_A \int b'} \ . \tag{8.31}$$

  Thus the braiding between axion string and the magentic surface defect is also non-Abelian.

- **Three-loop braiding correlation function.** The operator $e^{i\frac{q_m\alpha}{2\pi}\int d\theta}$ emitted from the braiding between the magnetic line defect and magnetic surface defect can further braid with axion string. Denote the axion string by $U_{q_A}$ that carries $\oint d\theta = 2\pi q_A$, supported on worldsheet $\Sigma_A$ bounded by volume $V_A$. Denote the worldsheet of the magnetic line $\mathcal{W}_{(0,q_m)}$ defect by $\Sigma_M$, and the surface that supports the magnetic surface defect $U_{E,\alpha}$ by $\Sigma_E$ that is bounded by volume $V_E$. Then the above braiding process gives the statistical correlation function

$$\langle U_{q_A}(\Sigma_A)U_{E,\alpha}(\Sigma_E)\mathcal{W}_{(0,q_m)}(\gamma_M)\rangle = e^{iq_A q_m \alpha \int \delta(V_A)^\perp \cup \delta(V_E)^\perp \cup \delta(\Sigma_M)^\perp} \ , \qquad (8.32)$$

where the integral in the exponent is the triple linking number of $\Sigma_A, \Sigma_E, \Sigma_M$. In other words, the axion string, worldsheet of magnetic lines, and the magnetic surface defects have three-loop braiding correlation function.

# Acknowledgement

The author thanks Maissam Barkeshli, Thomas Dumitrescu, Sergei Gukov, Anton Kapustin and Du Pei for conversations on related topics. The work is supported by the Simons Collaboration of Global Categorical Symmetry.

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
