# Peer review of "Non-Invertible Defects in Nonlinear Sigma Models and Coupling to Topological Orders"

_SciPost Physics_

## Round 2 · Referee Report · Anonymous (Referee 1) · 2023-6-6

Report

This paper has an extremely high scientific quality, discussing possible defect configurations arising in non-linear sigma models, possibly coupled to TQFTs. If the defect configurations discussed in the paper are completely topological, then one obtains interesting higher-group and non-invertible symmetry structures, whose exploration is an area of intense recent research. Non-linear sigma models are extremely important to understand from the point of view of understanding the infrared phases of interesting quantum field theories.

That being said, the writing of this paper is below-par. It seems like a set of personal notes for the author, rather than an effort to communicate with the wider academic audience. Even though the current structuring of the paper into the various sections is very nice, the structing within each section can be immensely improved.

A general comment is that there is often too much content in a single paragraph, which is not structured in a coherent way. In many cases, the crucial scientific details of the arguments are lacking, and the text reads like a compendium of results.

In more detail, I have the following comments:

  1. In the second paragraph of the introduction, we encounter the sentence "For instance, the correlation functions of the topological defects that generate one-form symmetry constrains confinement in gauge theories," I think the statement should be about the correlation functions of non-topological defects charged under the one-form symmetry. For example, the perimeter or area law exhibited by such correlation functions distinguishes deconfinement from confinement respectively.

  2. In the last bullet point of section 2.1.1, there are other references that should appear alongside [19] in the context of producing defects using gauged SPT phases. Exactly the same construction appeared before [19] in the paper $\href{https://arxiv.org/abs/2208.05973}{2208.05973}$. Actually the construction of 2208.05973 is more general, considering also defects arising from general gauged G-symmetric TQFTs that are not SPTs. In fact, going by the analogy with theta terms that the author points, such defects are now referred to as $\textit{theta defects}$ in the literature. These have been subsequently used in a lot of future developments in the field of generalized global symmetries: $\href{https://arxiv.org/abs/2212.06159}{2212.06159}$, $\href{https://arxiv.org/abs/2212.06842}{2212.06842}$, $\href{https://arxiv.org/abs/2212.07393}{2212.07393}$, $\href{https://arxiv.org/abs/2305.17159}{2305.17159}$.

  3. In point (1) appearing in section 2.2, it is not explained how the homeomorphism $\rho:~M\to M$ makes an appearance in the discussion of magnetic codimension one defects. According to the preceding discussion, such defects are characterized by maps $S^0\to M$, but it is not obvious how such maps are related to homeomorphisms $\rho$.

  4. Similar issues arise in the discussion of higher codimensional magnetic defects. In point (2) of section 2.2, it is claimed that magnetic defects of codimension two are characterized by conjugacy classes of $\pi_1(M)$. In this case, it is easier to see a connection between the general characterization of magnetic defects as maps $S^1\to M$ and $\pi_1(M)$, but it still leaves many things unexplained. The first question is what is the criterion the author uses to reduce the discussion from arbitrary maps $S^1\to M$ to equivalence classes of such maps characterized by $\pi_1(M)$. The second and more important question is why the discussion reduces only to conjugacy classes of $\pi_1(M)$. As is clear, there are many important details that are missing from the discussion at this point in the text.

  5. The point (4) is slightly confusing as well. The way I read the discussion is that for the codimension $k$ theta type magnetic defect to exist, the associated $(k-1)$-codimensional defect has to be characterized by a map $S^{k-2}\to M$ which is homotopic to the trivial map in which all of $S^{k-2}$ is mapped to the basepoint of $M$. Thus, it would seem that the $(k-1)$-codimensional defect has to be trivial except for $k=2$. This is because the existence of such a homotopy implies that the $(k-1)$-codimensional defect corresponds to the trivial element of $\pi_{k-2}(M)$. If this is indeed the case, the author should mention that. On the other hand, if this is not the case, the author should explain why not. The author does mention the existence of codimension three theta type magnetic defects in the following example $M=S^2$, so the latter option is more likely.

  6. In the example $M=S^1$ on page 7, the author should explain why the theta term type magnetic defects arise only for $U(1)\subset O(2)$ and not all of $O(2)$. Quite likely the explanation is related to the point I raised above, according to which we want the associated map $S^0\to M$ to be homotopic to the trivial map, which would imply that codimension two theta term type magnetic defects exists only for connected part of isometry group. But as I discussed above, this argument raises the question whether codimension bigger than two theta term type magnetic defects even exist.

  7. The author includes a $G$ gauge theory as an example of a non-linear sigma model. Although this is true, the paper discusses general aspects for the vanilla non-linear sigma models involving only scalar fields on spacetime. And these general aspects do not apply to gauge theories. For example, there is a claim at the bottom of page 5 that isometries of $M$ describe 0-form symmetries of the sigma model. I don't think that isometries of $BG$ are 0-form symmetries for gauge theory. This has to do with the fact that the discussion on page 5 assumes a particular form of the kinetic term, which is not applicable to the gauge theory case. However, this distinction has not been emphasized and explained in the paper. This can be extremely confusing for a reader wishing to apply general discussions to gauge theories. In this light, the author should either restructure the general discussions, or preface all general discussions related to isometries indicating that they won't be applicable to gauge theories.

  8. The second sentence of section 2.3 "When all these constituent defects are topological, the higher-codimensional junction is also topological." can be confusing. As the author already appreciates at other points in the paper, one can fuse such junctions with ambient non-topological defects, thus forming non-topological junctions. So just because the constituent defects are topological, there is no guarantee that there junctions will be topological. Thus, the author should rephrase the discussion here.

  9. I cannot parse equation (2.6): $f_k$ is a map from $S^{k-1}$ to Isom$(M)$ and $\rho$ is a map from Isom$(M)\times M$ to $M$. How is their composition $\tilde m_k$ a map from $S^{k-1}$ to $M$? The author should add some more details for clarity. More generally, there are many terse arguments scattered throughout the paper and a lot is demanded from the reader in being able to decode them. More explanations and clarifying remarks would immensely improve the quality of this work.

  10. Equation (3.1) discusses a $D$-dimensional electric "defect", where $D$ is the total spacetime dimension. This is a theta angle, and so is a term modifying the theory. Consequently the codimension one "defect" attached to this codimension zero electric "defect" is more appropriately an interface between two different $D$-dimensional sigma models having different topological actions $\omega$. This is an important remark to be made in the discussion of section 3.1.1 and similar discussions (e.g. section 4.1.5) that appear later in the paper, in order to avoid confusion.

  11. I believe there is an important typo right before equation (3.3). We should have $i_m^B\omega^{(D)}=\frac{2\pi}{N}y$ instead of $i_m^B\omega^{(D)}=\frac{2\pi}{N}dy$.

  12. At the beginning of section 3.2, it would be useful to remark that the discussion applies even if the topological term $\omega=0$.

  13. The discussion in section 3.3 regarding energy costs for moving around defects is confusing because we often hear that defects have "infinite tension". Consequently, the energy cost should be infinite. Perhaps, the author has in mind a situation where the defects in the IR sigma model arise from dynamical excitations having finite non-zero tension in a UV theory, and the author is then discussing the energy cost for moving the solutions containing these excitations around. The author should clarify the discussion of this section.

  14. At the bottom of section 6.1.1, the author introduces a terminology "condensation defect". However, the defects under discussion are not what have come to be referred to as condensation defects in the hep-th literature over the last year. For this reason, the discussion of condensation defects in this paper can be extremely confusing for readers. The prevalent definition of a condensation defect is that it is a topological codimension $k$ defect which admits a topological codimension $(k+1)$ defect along its boundary. The definition of condensation defect in section 6.1.1 seems to be a (possibly non-topological) defect appearing at the end of a topological defect. Although the two definitions seem related, they are quite different. As a consequence, I request the author to change the terminology and refer differently to the defects being currently referred to as condensation defects in the paper.

In conclusion, the paper is of extremely high quality, though needs a lot of improvement as far as presentation is concerned. I have made some concrete suggestions for improvement above, but I request the author to also revisit the general structuring of the paper in the light of these comments.

---

## Round 2 · Referee Report · Anonymous (Referee 2) · 2023-6-9

Report

The paper provides a systematic analysis of topological defects in generalised sigma models and applications to generalised symmetries. This includes generalised target spaces, including classifying spaces of Lie groups, and therefore also applies to gauge theories. This unified approach to sigma models and gauge theories is illuminating. The general analysis is hard to follow in places for this reviewer but this is remedied by the many examples that are presented. I am therefore happy to recommend for publication. I have the following suggestions for improvement:

  • In footnote 2, could the author state explicitly that those defects in class (2) that coincide with defects in (1) are those in the -- image -- of the connecting map, if that is the correct statement?
  • Page 6, item (2). Could the author explain briefly why conjugacy classes in the homotopy group is the correct notion? Presumably, those in the center are topological? If so, perhaps that could be stated? This should reproduce the classification of Gukov-Witten defects for the classifying space of a connected Lie group.
  • Page 6, could the author clarify what is mean by an "integral" element in the homotopy group?
  • I found the general discussion at the beginning of 2.3.1 hard to follow without skipping ahead to the examples. For example, it wasn't clear to the reviewer what "has a boundary condition with winding number" means without skipping ahead. One recommendation here: the more explicit description of tri-valent junctions at the beginning of 2.3.2 was very clear and could be moved up to 2.3.1 with SO(3) -> Isom(M) to make the discussion easier to follow there. Is it possible to describe co-dimension three junctions so explicitly?
  • Section 3.1.1: it is said an isometry induces a "permutation action" on topological actions. Is there an assumption of a finite or disconnected group of isometries? Couldn't this act on representatives i.e. the form $\omega$ by pull-back but leave the class $[\omega]$ invariant? Below (3.1) one could mention this is a domain wall. Section 7.1.2: could the author clarify why the denominator is $U(N_f-1)$ and not $SU(N_f-1)$? What then is the quotient - not a smooth projective space?

I also notices the following typos: - Abstract: "criticality admit" -> "criticality that admit". - First line: "context" -> "contexts". - Beginning of section 2.2.1: "electric" -> "magnetic".

---

## Editorial Decision

awaiting_resubmission